# RePlan: Reasoning-Guided Region Planning for Complex Instruction-Based Image Editing

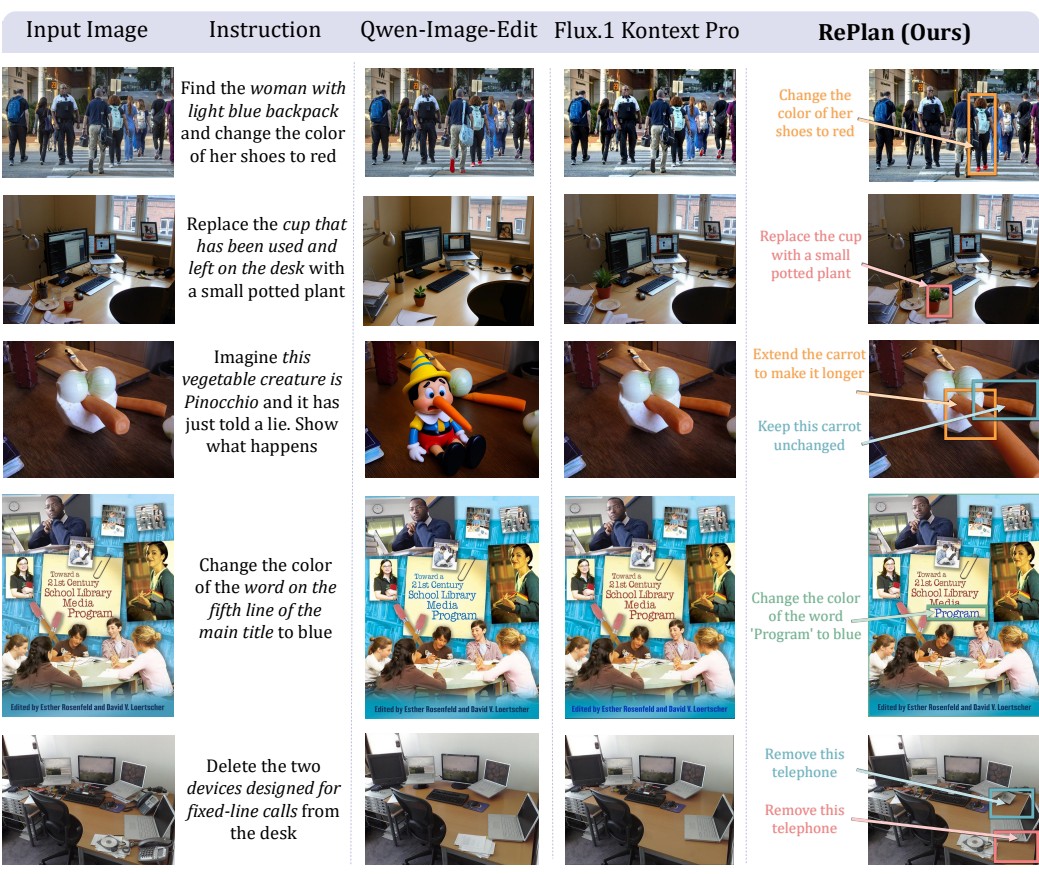

Figure 1: We define **Instruction-Visual (IV) Complexity** as the challenges that arise from complex input images, intricate instructions, and their interactions—for example, cluttered layouts, fine-grained referring, and knowledge-based reasoning. Such tasks require models to conduct fine-grained visual reasoning. To address this, we propose **RePlan**, a framework that leverages the inherent visual understanding and reasoning capabilities of pretrained VLMs to provide region-aligned guidance for diffusion editing model, producing more accurate edits with fewer artifacts than open-source SoTA baselines (Batifol et al., 2025; Wu et al., 2025a). Zoom in for better view.

## ABSTRACT

Instruction-based image editing enables natural-language control over visual modifications, yet existing models falter under Instruction–Visual Complexity (IV-Complexity), where intricate instructions meet cluttered or ambiguous scenes. We introduce RePlan (Region-aligned Planning), a plan-then-execute framework that couples a vision–language planner with a diffusion editor. The planner decomposes instructions via step-by-step reasoning and explicitly grounds them to target regions; the editor then applies changes using a training-free attention-region injection mechanism, enabling precise, parallel multi-region edits without iterative inpainting. To strengthen planning, we apply GRPO-based reinforcement learn-

ing using 1K instruction-only examples, yielding substantial gains in reasoning fidelity and format reliability. We further present IV-Edit, a benchmark focused on fine-grained grounding and knowledge-intensive edits. Across IV-Complex settings, RePlan consistently outperforms strong baselines trained on far larger datasets, improving regional precision and overall fidelity. We will release all our codes and data soon.

# 1 INTRODUCTION

Instruction-based image editing has emerged as a core direction in multimodal AI, enabling users to flexibly modify images through natural language. Existing pure editing models (Zhang et al., 2023; Liu et al., 2025a; Brooks et al., 2023; Batifol et al., 2025) already produce diverse and high-quality visual effects, yet they still struggle with accurately grounding and executing edits within visually and linguistically complex scenarios, a challenge we formalize as **Instruction-Visual Complexity (IV-Complexity)**.

We define IV-Complexity as the intrinsic challenge that arises from the interplay between visual complexity, such as cluttered layouts or multiple similar objects, and instructional complexity, such as multi-object references, implicit semantics, or the need for world knowledge and causal reasoning. The interaction between these dimensions even amplify the challenge, requiring precise editing grounded in fine-grained visual understanding and reasoning over complex instructions. As the second row shown in Figure 1, in a cluttered desk scene, the instruction "Replace the cup that has been used and left on the desk with a small potted plant" requires distinguishing the intended target among multiple similar objects and reasoning about implicit semantics that what counts as a "used" cup. This combined demand illustrates how IV-Complexity emerges when visual and instructional factors reinforce each other.

Recent progress in large-scale vision-language models (VLMs) (Wang et al., 2024; Bai et al., 2025; Chen et al., 2024b;a; Lai et al., 2024; Liu et al., 2025c) has demonstrated strong capabilities in visual understanding and world-knowledge reasoning. A natural idea is therefore to transfer these strengths into instruction-based editing. Inspired by this, methods such as Qwen-Image (Wu et al., 2025a), Bagel (Deng et al., 2025), and UniWorld (Lin et al., 2025) attempt to unify VLMs with image generation models, showing remarkable potential. However, these unified approaches typically treat VLMs as semantic-level guidance encoders, which leads to coarse interaction with the generation model . As a consequence, even with massive training data, they still lag behind the fine-grained grounding and reasoning abilities that standalone VLMs can achieve. For example, while a VLM can correctly localize targets in a complex grounding task (Lin et al., 2014; Yu et al., 2016), the corresponding editing model may fail to identify the same regions for modification under similar instructions.

We therefore pose the question of how the fine-grained perception and reasoning capacities of VLMs can be more effectively exploited to overcome IV-Complexity in image editing. Our key insight is that the interaction between VLMs and diffusion models should be refined **from a global semantic level to a region-specific level.** Rather than using VLMs merely as high-level semantic encoders, we harness their fine-grained perception and reasoning capabilities to generate region-aligned guidance that explicitly links decomposed instructions to target regions in the image. Building on this insight, we propose RePlan, a framework that couples VLMs with a diffusion-based decoder in a plan–execute manner: the VLM performs chain-of-thought reasoning to analyze the visual input and instruction, outputs structured region-aligned guidance, and the diffusion model faithfully executes this guidance to complete precise edits under IV-Complexity.

To accurately ground edits to the regions specified by the guidance, we propose a **training-free attention region injection** mechanism, which equips the pre-trained editing DiT (Batifol et al., 2025) with precise region-aligned control and allows efficient execution across multiple regions in one pass. This avoids the image degradation issues of multi-round inpainting while reducing computation cost, offering a new perspective for controllable interactive editing. On top of this framework, we further enhance the planning ability of VLMs through GRPO reinforcement learning. Remarkably, with only ∼1k instruction-only examples, RePlan outperforms models trained on massive-scale data and computation when evaluated under IV-Complex editing task.

However, existing instruction-based editing benchmarks (Ye et al., 2025; Liu et al., 2025a) oversimplify editing scenarios by emphasizing images with salient objects and straightforward instructions. Such settings fail to reflect the real-world challenges and diverse user needs posed by IV-Complexity. To bridge this gap, we introduce **IV-Edit**, a benchmark specifically designed to evaluate instruction–visual understanding, fine-grained target localization, and knowledge-intensive reasoning.

We summarized our contributions as:

- **RePlan Framework:** We propose RePlan, which refines VLM–diffusion interaction to region-level guidance. With GRPO training from only a small set of instruction-only examples, RePlan outperforms state-of-the-art models trained on orders of magnitude more data in IV-Complexity scenarios.
- **Attention Region Injection:** We design a training-free mechanism that enables accurate response to region-aligned guidance, while supporting multiple edits in one-pass.
- **IV-Edit Benchmark:** We establish IV-EDIT, the first benchmark tailored to IV-Complexity, providing a principled testbed for future research.

## 2 RELATED WORK

**Instruction-Based Image Editing**. Instruction-driven image editing has advanced with diffusion-based methods. End-to-end approaches such as *InstructPix2Pix* (Brooks et al., 2023; Hui et al., 2024) learn direct mappings from instructions to edited outputs, showing strong global editing but limited spatial reasoning. Inpainting-based pipelines first localize regions and then apply mask-guided editing (Zhang et al., 2023), which improves locality but depends on fragile localization modules and struggles with reasoning-heavy instructions. More recent lines explore VLM-guided generation (Wu et al., 2025a; Deng et al., 2025), but typically leverage VLMs only at a coarse level, underutilizing their fine-grained reasoning capabilities.

**Vision–Language Models**. Large VLMs (Wang et al., 2024; Bai et al., 2025; Chen et al., 2024b) exhibit remarkable fine-grained perception (Lai et al., 2024; Wang et al., 2024) and complex reasoning abilities (Liu et al., 2025b; Goodfellow et al., 2016). These strengths suggest great potential for boosting IV-Complex image editing.

**Image Editing Benchmarks**. Existing benchmarks such as Imgedit (Ye et al., 2025) and GEdit (Liu et al., 2025a) mainly evaluate edits on images with clean layouts and explicit instructions. Reasoning-oriented benchmarks like *KrisBench* (Wu et al., 2025b) and *RISEBench* (Zhao et al., 2025) move beyond direct commands, but their tasks still involve simple image compositions and fail to reflect the intertwined linguistic-visual complexity of real-world editing.

## 3 METHOD

### 3.1 OVERVIEW

We present a framework for complex instruction-based image editing that couples a vision–language model (VLM) with a diffusion-based decoder. The VLM interprets the input image and instruction, conducts chain-of-thought reasoning, and outputs region-aligned guidance. This guidance is executed by a DiT decoder through a training-free attention region injection, enabling one-pass, multi-region editing. The overall framework is illustrated in Figure 2. To further strengthen the planner, we apply reinforcement learning with VLM-based feedback on post-edited results, achieving significant gains with only $\sim$1k instruction-only samples, without requiring paired images.

### 3.2 REGION-ALIGNED EDITING PLANNER

**Reasoning on Instructions**. Given an input image $I \in \mathbb{R}^{H \times W \times 3}$ and a user instruction $\mathcal{T}$, the VLM planner first reasons about editing targets by combining image understanding with instruction analysis. For ambiguous or abstract descriptions, the planner must ground high-level semantics into concrete visual effects. We further enhance this reasoning ability using the GRPO reinforcement learning algorithm (see Section 3.4).

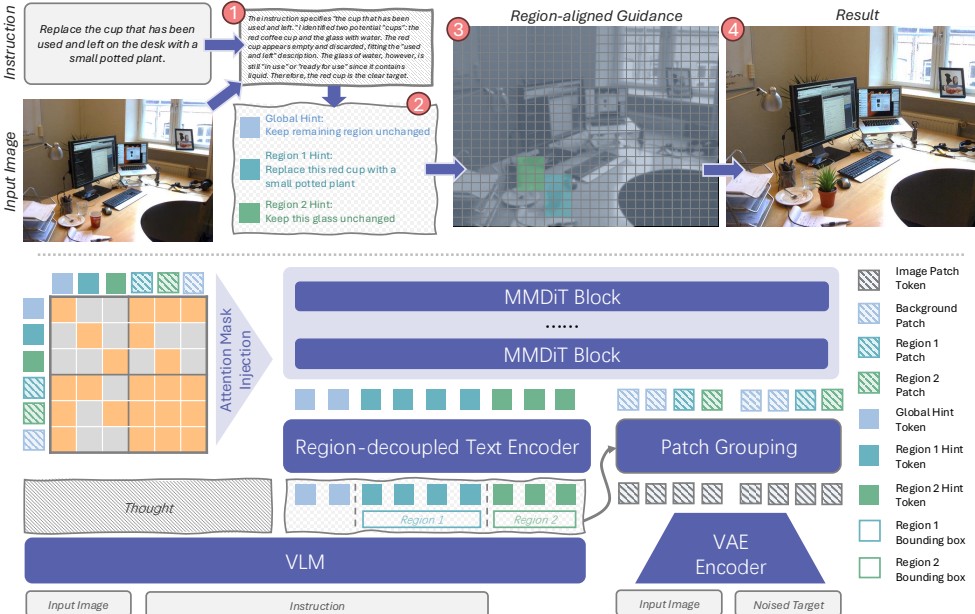

Figure 2: **Overview of our RePlan framework.** The bottom part of the figure shows the overall architecture. Given an input image and text instruction, the VLM analyzes them via chain-of-thought reasoning and produces region-aligned guidance, where each guidance includes a region bbox and its editing hint. Each hint is futher encoded by a text encoder into a feature token, while image patch tokens are obtained by VAE encoding and grouped according to the region bounding boxes. A group-specific attention mechanism, detailed in Figure 4, is proposed to allow MMDiT to generate the final edited image. The top part of the figure presents an editing examples.

**Region-aligned Editing Planning**. We decouple the editing guidance according to the target into global edits and regional edits. We represent all editing guidance as region–hint pairs:

$$\{(B_k, h_k)\}_{k=0}^K,$$

where $B_0$ denotes the entire image (for global edits) with its associated hint $h_0$ (e.g., style or background adjustment), and $B_k$ ($k \geq 1$) are bounding boxes for local edits with corresponding hints $h_k$. Hints can also be negative instructing that a region remain unchanged, which helps prevent editing effects from unintentionally bleeding into neighboring areas.

> **\<think\>** *The instruction specifies "the cup that has been used and left." I identified two potential "cups": the red coffee cup and the glass with water. The red cup appears empty and discarded, fitting the "used and left" description. The glass of water, however, is still "in use" or "ready for use" since it contains liquid. Therefore, the red cup is the clear target.* **\</think\> \<global\>** Keep remaining region unchanged **\</global\> \<region\>[** {"bbox_2d": [224, 372, 263, 431], "hint": "Replace this red cup with a small potted plant"}, {"bbox_2d": [175, 329, 220, 388], "hint": "Keep this glass unchanged"} **]\</region\>**

Figure 3: VLM output format Example

**Output Format**. We require the VLM to output structured text for convenient post-processing, with explicit markers separating reasoning, global edits, and region guidance. Region guidance are expressed in JSON format. An example is shown in Figure 3.

**Interactivity**. Explicit region planning enhances interpretability and controllability. When the automatically generated guidance is insufficient, users can adjust regions or the associated hints directly.

## 3.3 TRAINING-FREE ATTENTION REGION INJECTION

**Preliminary**. Our method builds upon the MMDiT (Multimodal Diffusion Transformer) (Esser et al., 2024; Batifol et al., 2025) framework for instruction-based image editing. MMDiT concatenates text, image, and latent tokens into a single sequence, which is processed jointly by Transformer self-attention, enabling rich cross-modal interaction without introducing extra modules.

The input consists of two modalities: the editing instructions $\mathcal{T}$, the original image $I$. They are embedded as

$$F^{\text{text}} = E_{\text{text}}(\mathcal{T}), \quad F^{\text{img}} = E_{\text{img}}(I). \tag{1}$$

The embeddings are concatenated into a unified input sequence:

$$X^{(0)} = [F^{\text{text}} \,\|\, F^{\text{img}} \,\|\, \mathbf{z}_t].\tag{2}$$

Where $\mathbf{z}_t$ is the noised latentat step $t$. While full self-attention allows free information exchange across modalities, it also causes interference in multi-region editing: tokens from one region may attend to unrelated instructions, leading to *target confusion* or *instruction failure*.

**Text encoding and grouping**. We split the editing hints into one global hint $h_0$ associated with the full image $B_0$, and $K$ local hints $\{h_k\}_{k=1}^{K}$ each associated with a bounding box $B_k$. Hints are separately encoded and concatenated as

$$F^{\text{text}} = [\, E_{\text{text}}(h_0) \,\|\, E_{\text{text}}(h_1) \,\|\, \cdots \,\|\, E_{\text{text}}(h_K) \,].\tag{3}$$

We thus define token index groups $G_0^{\text{text}}, \ldots, G_K^{\text{text}}$ accordingly.

**Image encoding and patch grouping**. The image $I$ is first processed by the VAE encoder to produce a spatial feature map $F^{\text{img}}$ which can be reshaped into patch tokens $\{f_{i,j}\}_{i=1..M, j=1..N}$. Each editing region $B_k$ is then mapped into the patch grid to collect the corresponding group:

$$G_k^{\text{img}} = \{f_{i,j} \mid (i,j) \in B_k\}, \qquad k = 1, \ldots, K,\tag{4}$$

while the background group is defined as patches not belong to any region groups:

$$G_{\text{bg}}^{\text{img}} = \{f_{i,j}\}_{i,j} \setminus \bigcup_{k=1}^{K} G_k^{\text{img}}.\tag{5}$$

**Attention mask manipulation**. In each attention layer, we impose a binary mask $M \in \{0,1\}^{|X|\times|X|}$ controlling which tokens can attend to which others. The mask follows five intuitive rules, as also visualized in Figure 4:

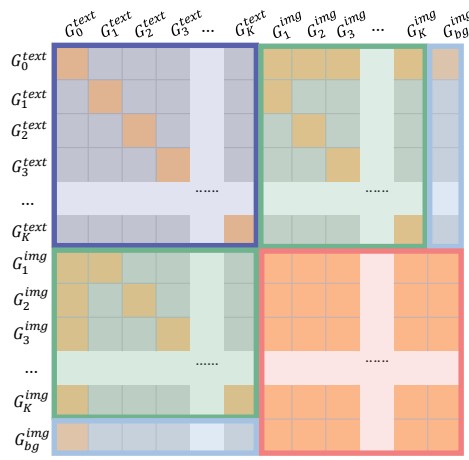

Figure 4: Attention rule visualization. We use different highlight colors to indicate different rules, which correspond to Hint isolation, Region constraint, Background constraint and Image–latent full interaction.

1. **Intra-group interaction.** Tokens within the same group (text, image, or latent) are fully connected. This ensures that local context is preserved inside each modality or region.

2. **Hint isolation.** Different text groups $G_a^{\text{text}}$ and $G_b^{\text{text}}$ ($a \neq b$) are not allowed to see each other. This prevents regional instructions from contaminating one another and avoids semantic conflicts.

3. **Image–latent full interaction.** All image and latent tokens remain globally connected. This ensuring global stylistic coherence and smooth boundaries. Meanwhile, the effect can extend beyond the bounding box when necessary.

4. **Region constraint.** Tokens belonging to region $B_k$ ($u \in G_k^{\text{img}}$) may only attend to their own hint tokens $G_k^{\text{text}}$ and the global instruction $G_0^{\text{text}}$. In this way, local edits are precisely guided by their designated hints while still aligned with the global change.

5. **Background constraint.** Background tokens $u \in G_{\text{bg}}^{\text{img}}$ can only attend to the global text group $G_0^{\text{text}}$. This keeps the untouched background in sync with the global instruction without being polluted by local edits.

Together, these rules ensure that (i) text instructions are disentangled, (ii) image spaces preserve global coherence, and (iii) each regional hint remains focused on its designated area. As a result, MMDiT can execute multiple region-level edits in parallel, enhancing efficiency while avoiding the accumulated errors of multi-round inpainting, and further supports region-level negative prompts.

### 3.4 STRUCTURED PLANNING AND REASONING WITH GRPO

We perform reinforcement learning on the pretrained vision–language model (VLM) to improve its planning capabilities for complex instruction-based editing. Specifically, we use Qwen2.5-VL 7B (Bai et al., 2025) as the VLM planner, and Flux Kontext Dev (Batifol et al., 2025) as the diffusion image decoder.

We adopt Group Relative Policy Optimization (GRPO) (Shao et al., 2024), which updates the planner by comparing the relative quality of multiple edited outputs generated from the same instruction.

Since rewards rely on valid image outputs, errors in plan formatting can disrupt decoding and lead to distorted reward signals. To address this, we employ a two-stage training strategy: we first focus on improving plan validity and reasoning quality, and then introduce image-level rewards to refine planning behavior.

Both stages use only 1k complex instruction editing samples we generated for supervised alignment before reinforcement learning.

**Stage 1: Format and reasoning learning**. In the first stage, GRPO training provides only format-related rewards to ensure structured plan generation and coherent reasoning:

- **Tag format reward.** A regular expression parser checks whether the output follows the tag structure in Figure 3. A valid structure yields a positive reward; otherwise zero.
- **Region format reward.** The content inside the `<region>` tag is parsed as JSON, including the outer list and each inner dictionary. Valid JSON yields a positive reward; otherwise zero.
- **Reasoning quality reward.** The length of the content inside the `<think>` tag is measured, and the reward increases with length from zero up to a capped maximum.

The Stage 1 reward can be computed as:

$$R^{(1)} = R_{Ta} + R_F + R_R.$$

**Stage 2: Planning learning**. In the second stage, plans are decoded into images, and a larger VLM provides image-level evaluation. We adopt Qwen2.5-VL 72B as the reward model:

- **Target** ($R_T$): Whether the edit is applied to the specified area.
- **Effect** ($R_E$): Whether the visual change matches the instruction.
- **Consistency** ($R_C$): Reservation of irrelevant regions and global style.

To prevent reward hacking (e.g., maximizing consistency by making no edits), consistency is re-weighted by effect: $R'_C = R_C \cdot R_E$. Finally, the Stage 2 reward is:

$$R^{(2)} = R_T + R_E + R'_C + \lambda R^{(1)},$$

where $\lambda$ is a small weight that preserves format reliability.

## 4 DATA CONSTRUCTION AND BENCHMARK

**Task Setting**. IV-Complexity highlights the inherent difficulty of faithfully grounding user intent within rich visual contexts, where instructions often involve complex referring expressions and require fine-grained reasoning to align language with specific visual regions. Motivated by these challenges, we design the IV-Edit Benchmark around two representative task scenarios: (1) complex real-world photo editing and (2) text-related image editing. In both scenarios, we deliberately emphasize images with diverse, non subject-dominated content and editing instructions that demand detailed visual understanding, often combined with world knowledge reasoning. This setting reflects the essence of IV-Complexity, providing a challenging testbed for instruction-based image editing.

Specifically, each instruction is structured into a reference expression and an editing task. We consider a total of 7 referring types and 16 task types, as illustrated in Figure 5a and Figure 5b. Full definitions of reference categories and task types are provided in Appendix B.

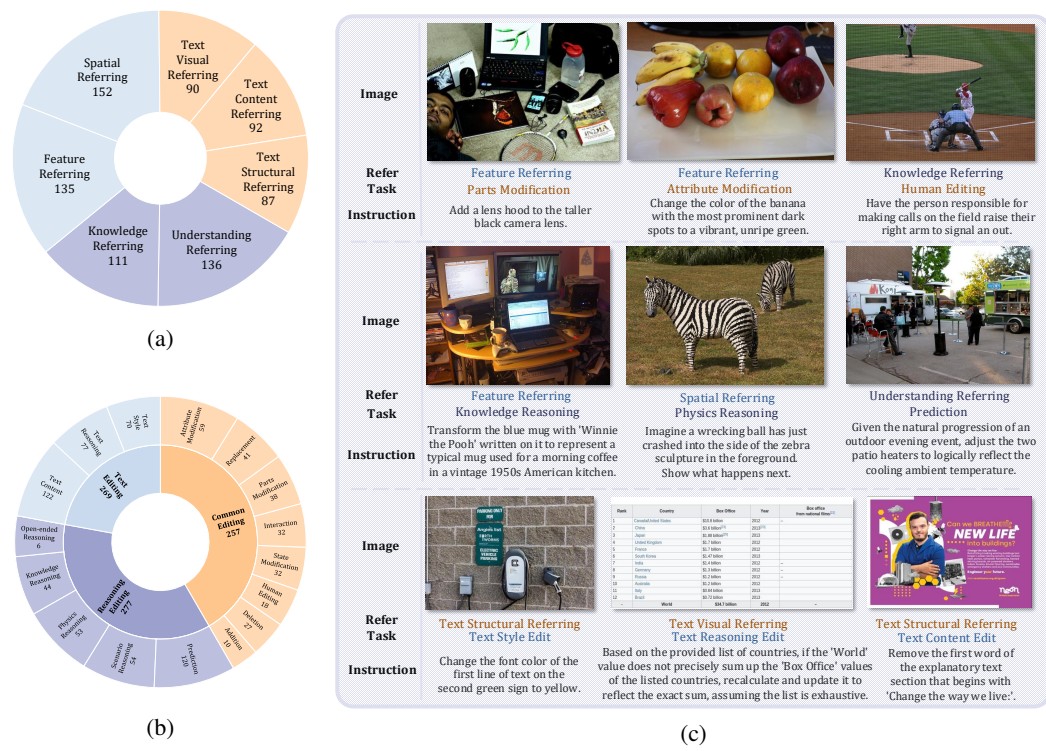

(a)

(b)

(c)

Figure 5: **Overview of our IV-Edit Benchmark.** (a) and (b) respectively shows the distribution of referring types and task types across the dataset. IV-Edit is explicitly designed to reflect the IV-Complexity challenge, where user instructions require aligning fine-grained language with rich and diverse visual contexts. (c) presents visual examples spanning a wide range of real-world scenarios and fine-grained instruction intents—including spatial, structural, and reasoning-intensive edits. Each instruction is decomposed into a referring expression and a task type, reflecting the need for both grounded understanding and visual transformation.

**Benchmark Statistics**. With careful filtering and manual verification, our IV-Edit benchmark comprises around 800 instruction–image pairs. On average, the instructions contain 21 words, and 182 examples involve edits across multiple target regions. The distribution of referring expressions and editing task categories is summarized in Figure 5a and Figure 5b, while Figure 5c illustrates representative cases of IV-Edit. Additional dataset statistics are provided in Appendix B.

**Evaluation Protocol**. Since the input images are not dominated by a single subject, traditional metrics that compute global semantic similarity based on CLIP are not well suited. We introduce Gemini2.5-Pro as a fine-grained evaluator, assigning 5-point ratings to image pairs before and after editing along the following four dimensions:

- **Target**: Whether the target region is correctly located and modified, avoiding confusion between editing objects. When multiple different targets need to be edited, whether there are no omissions.

- **Consistency**: Whether the content outside the edited region remains unchanged, without editing effects spilling into unrelated areas. In addition, whether the overall style before and after editing remains stable.

- **Quality**: The visual quality of the editing results. Regardless of the editing instructions, whether the edited image contains artifacts or style conflicts between different regions.

- **Effect**: Whether the visual effect of the editing instruction is accurately achieved.

Table 1: Quantitative comparison of open-source and proprietary image editing models on four evaluation dimensions. We also report Overall and Weighted scores. For open-source models, the highest score in each column is marked as **Bold**, while the second highest is indicated with Underline. RePlan achieves the best consistency and overall score among open-source models.

| Model | Quality ↑ | Target ↑ | Effect ↑ | Consistency ↑ | Overall ↑ | Weighted ↑ |
|---|---|---|---|---|---|---|
| Gemini-Flash-Image | 3.89 | 4.11 | 3.93 | 2.89 | 3.71 | 3.44 |
| GPT-4o | 3.61 | 4.02 | 3.78 | 1.77 | 3.30 | 3.07 |
| InstructPix2Pix | 2.47 | 2.47 | 1.90 | 1.40 | 2.06 | 1.48 |
| Uniworld-V1 | 3.26 | 2.89 | 2.18 | 1.46 | 2.45 | 1.84 |
| Bagel-Think | 3.44 | 3.47 | 2.93 | 2.33 | 3.05 | 2.46 |
| Qwen-Image | 3.47 | 3.72 | **3.24** | 1.79 | 3.05 | 2.62 |
| Flux.1 Kontext Dev | 3.93 | 3.34 | 2.73 | 2.88 | 3.22 | 2.49 |
| **RePlan (Flux.1 Kontext)** | **4.16** | 3.47 | 2.59 | **3.64** | 3.46 | 2.55 |
| **RePlan (Qwen-Image)** | 3.86 | **3.77** | 3.16 | 3.24 | **3.51** | **2.91** |

## 5 EXPERIMENTS

### 5.1 RESULTS ON VI-EDIT BENCHMARK

**Metric**. We evaluate along four dimensions from Section 4 using Gemini-2.5-Pro. The Overall score is the simple average of these dimensions. To avoid inflated Consistency when no edits are made, we introduce a Weighted score that weights Consistency by Effect, defined as Weighted $= \sum_{\text{samples}} (\text{Target} + \text{Quality} + \text{Effect} + \text{Effect} \times \text{Consistency})/4$, with all scores ranging from 1 to 5.

**Evaluation Setting**. We conduct evaluations on a total of two closed source models and six open source models. The closed source models include GPT-4o and Gemini-2.5-Flash-Image (also referred to as nano banana). The open source models evaluated are InstructPix2Pix (Brooks et al., 2023), Uniworld (Lin et al., 2025), Bagel (Deng et al., 2025)(using the think mode), Qwen-Image (Wu et al., 2025a), Flux.1 Kontext dev (Batifol et al., 2025), and our proposed RePlan. For our RePlan framework, we conduct evaluations by applying it to Flux.1 Kontext dev and Qwen-Image-Edit, both of which share the MMDiT architecture.

**Quantitative Analysis**. Table 1 reports the evaluation results. The accuracy of handling referring expressions is measured from two perspectives: Target and Consistency. Target emphasizes recall, capturing the model's ability to semantically localize the editing object, while Consistency emphasizes precision, reflecting fine-grained localization at the regional level. For Target, Qwen-Image and Bagel perform strongly by leveraging VLMs, which better resolve complex semantic referring and the underlying intentions in instructions. Regardless of whether Flux.1 Kontext dev or Qwen-Image-Edit is used as the MMDiT backbone, RePlan shows a significant performance improvement, highlighting its superior reasoning capability in analyzing and interpreting IV-complex instructions.

RePlan also achieves a clear advantage in Consistency, benefiting from directional regional injection that prevents editing spillover into semantically similar region, a common drawback of semantic-level guidance methods.

**Qualitative Analysis**. By comparing the editing results in Figure 6, we observe that our RePlan demonstrates clear advantages in accurately localizing the target editing regions. In contrast, other existing methods tend to suffer from editing spillover into semantically similar areas, a problem that persists even in state-of-the-art proprietary models. In addition, RePlan shows stronger reasoning ability for handling indirect instructions; for example, in the third case it not only identifies the word "June" in the image but also infers that the next month is "July." More comparative results can be found in Appendix H

**Ablation on Planner**. We test RePlan on IV-Edit using Gemini2.5-Pro and Qwen2.5-VL (Bai et al., 2025) as planners without RL. As shown in Table 4, both lag behind the RL-trained planner. Manual inspection reveals that Gemini2.5-Pro, though strong in reasoning, often produces bbox errors, while Qwen2.5-VL struggles with hint decomposition and format compliance. These results highlight the necessity of RL for a reliable planner.

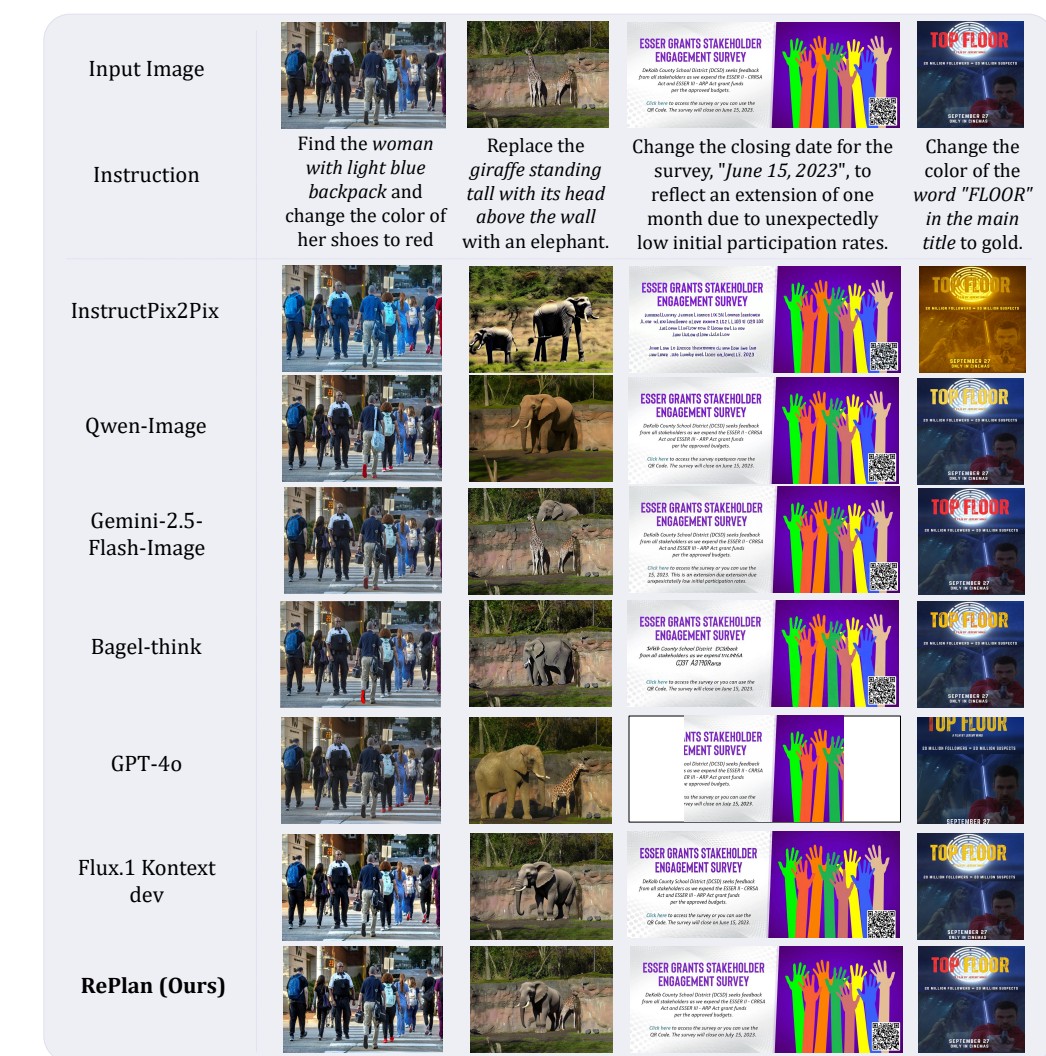

Figure 6: **Editing results comparison.** Notably, GPT-4o enforces fixed aspect ratios, leading to unavoidable cropping for non-standard images.

Table 2: **Comparison on the choice of zero-shot VLM region planner.** Flux.1 Kontext dev as the MMDiT backbone.

| Model | Overall ↑ | Weighted ↑ |
|---|---|---|
| Gemini2.5-pro | 2.95 (-0.51) | 1.93 (-0.62) |
| Qwen2.5-VL 7B | 2.60 (-0.86) | 1.63 (-0.92) |
| RePlan (Kontext) | 3.46 | 2.55 |

Table 3: **Ablation on reasoning and staged RL training strategy.** Flux.1 Kontext dev as the MMDiT backbone.

| Model | Overall ↑ | Weighted ↑ |
|---|---|---|
| w/o reasoning | 3.31 (-0.15) | 2.49 (-0.06) |
| Uni Stage RL | 3.42 (-0.04) | 2.51 (-0.04) |
| RePlan (Kontext) | 3.46 | 2.55 |

**Ablation on Reasoning**. To assess the role of CoT reasoning, we remove it and train the VLM to directly output region-aligned guidance (Table 3). Performance drops markedly without reasoning. Combined with the planner ablation, this underscores its importance in analyzing instructions and producing effective guidance.

**Ablation on RL Stage**. We further evaluate the two-stage RL strategy (Section 3.4) by skipping the first-stage format learning. Results show that the full two-stage scheme not only achieves higher final scores under the same training steps, but also delivers superior sample efficiency. This validates both the effectiveness and efficiency of the strategy.

## 6 CONCLUSION

We introduce Instruction–Visual Complexity (IV-Complexity), a new challenge from cluttered visuals and ambiguous instructions. Existing methods depend on coarse semantic guidance, limiting fine-grained control. To address this, we propose RePlan, which uses VLM-based region reasoning with diffusion models and a training-free attention injection for precise parallel edits. We also release IV-Edit, the first benchmark for IV-Complexity, providing a principled testbed for real-world instruction-based editing.

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

## A    APPEAL OF USING LLMS

We use LLM for polishing writing.

## B    MORE IV-EDIT STATISTICS

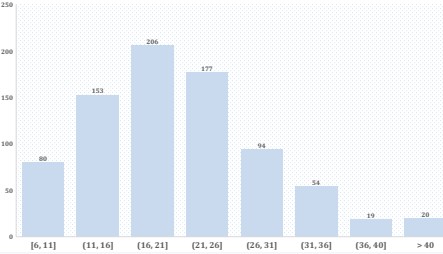

Figure 7: Instruction length distribution

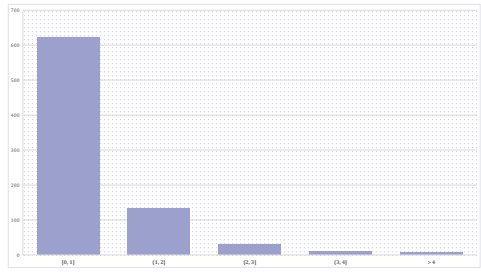

Figure 8: Distribution of expected editing region counts per instruction

**General Statistics of IV-Edit**. In Figure 7, we show the distribution of the total number of words in all instructions, with an average length of 21 words. In Figure 8, we present the distribution of the expected number of independent editing regions per instruction.

**Definition of Referring Types**. For text editing tasks, the referring types are defined as:

- Visual (90 samples): Locating a text element based on its visual design attributes, such as font, size, color, weight, or style. The focus is on the *appearance* of the text.
- Structural (87 samples): Locating a text element according to its logical position or role within the overall document structure or layout. The emphasis is on the element's *hierarchical position* in the document (e.g., heading, paragraph, list item).
- Content (92 samples): Locating a text element by referencing its exact content, partial content, or semantic meaning. The emphasis is on *what the text actually says*.

And for realistic image editing:

- Feature (135 samples): Locating an object directly by its objective, observable visual attributes, such as color, texture, material, pattern, size, shape, or state. This relies solely on information immediately visible in the image.
- Spatial (152 samples): Locating an object by its position in the scene, either in terms of its absolute location relative to the image frame (e.g., "top-left corner") or its relative position with respect to other objects in the scene (e.g., "beside the tree").
- Knowledge (111 samples): Locating an object by applying external, real-world knowledge that extends beyond the visual information contained in the scene. This includes object categories, functions, or cultural symbolism.
- Understanding (136 samples): Locating an object by inferring from contextual cues, behaviors, and relationships within the image. This requires deriving information that is implied but not explicitly depicted, such as intentions, emotional states, social roles, or causal relations.

**Definition ofTask Types**. Common image-based edits include:

- Add (10 samples): Introduce new objects or features realistically regarding lighting, perspective, and scale.
- Delete (27 samples): Remove specified target(s) completely and convincingly fill the space through inpainting.
- Replacement (41 samples): Substitute the specified object(s) with entirely different ones.

- Attribute (59 samples): Modify visual properties such as color, texture, material, brightness, or size.
- Parts Modification (38 samples): Add, remove, or alter specific parts of an object.
- State Modification (32 samples): Change the state or implied action of an object (e.g., closed book to open).
- Modify Human Animal (18 samples): Alter the appearance, pose, action, or clothing of a human or animal subject.
- Interaction (32 samples): Change interactions either between multiple targets (e.g., swap positions, face each other) or between target(s) and their environment (e.g., make a person hold an umbrella).

Reasoning-related tasks, including prediction-based edits, are as follows:

- Prediction (120 samples): Perform prediction-based edits.
    - Temporal: Predict plausible future states (e.g., show how ice cream melts over time).
    - Causal: Depict likely consequences of actions or events (e.g., what happens if a vase falls).
    - Logic: Resolve inconsistencies or complete logical patterns (e.g., make lighting consistent with shadows).
- Physics Reasoning (53 samples): Simulate the influence of physical or environmental conditions (e.g., strong wind affecting hair and clothes).
- Scenario Reasoning (54 samples): Imagine new events or scenarios, modifying targets or environments accordingly (e.g., a kitchen after a large dinner party).
- Open-Ended Reasoning (6 samples): Creative reasoning-based edits driven by "what if" narratives (e.g., two people secretly being agents in a café).
- Knowledge Reasoning (44 samples): Apply real-world or domain knowledge to edit (e.g., turning a building into the Eiffel Tower, dressing someone as a firefighter).

Text-related tasks include:

- Text Content Edit (122 samples): Modify textual content such as correcting typos, replacing words, updating information, or adding/deleting text elements.
- Text Style Edit (70 samples): Modify the visual properties of text such as font, size, color, style, or alignment.
- Text Reasoning Edit (77 samples): Generate or modify text based on logical or contextual reasoning (e.g., automatically calculating and filling in table values).

## C  DATA CONSTRUCTION DETAILS

Our train/test data construction pipelineis as follows:

1. **Source:** We used COCO, LISA ReasonSeg, TextSceneHQ split of Text Atlas, TableVQA-Bench (test set only), and TableQA as image sources.
2. **Image filter:** We used Gemini 2.5 Pro together with the datasets' own annotations to select images that fit the IV-Edit setting:
   - Complex scenes that are not subject-dominated, with multiple instances of the same category and clear content.
   - Or, for text-editing tasks, clearly structured charts and tables, and photos/slides/posters with multiple clear textual regions.
3. **Instruction construction:** For each sample, we randomly selected three candidate options from pre-defined referring categories and task categories. Then, using Gemini 2.5 Pro, we prompted the model to choose any reasonable combination based on the image content, generate the editing target referring, and finally construct the corresponding editing instruction. The expected number of referring target instances/regions was also randomly provided through prompting.

4. **Instruction filter:** We used Qwen2.5-VL-72B to annotate the bounding boxes (bbox) of all referring targets, first filtering out samples whose bbox count did not match the assigned target count. Next, we employed Gemini 2.5 Pro again to filter out samples with ambiguous referring targets or instructions that were difficult to realize through visual editing effects.

The data generated from the training splits of the source datasets were used as the training set. After filtering, the retained portion accounted for roughly one-third of the total source data. For the test set, we further applied manual sample-by-sample filtering.

## D   COMPARISON WITH GLOBAL REPHRASE

| Model | Consistency ↑ | Overall ↑ | Weighted ↑ |
|---|---|---|---|
| Gemini-2.5-Pro | 2.61 | 3.23 | 2.71 |
| Qwen2.5-VL 7B | 2.42 | 3.08 | 2.50 |
| Ours (Kontext) | 3.64 | 3.46 | 2.55 |

Table 4: Comparison with global instruction rephrase

We further compared the approach of using the VLM to perform only global instruction rephrasing, and then providing the rephrased prompt to flux.1 kongtext dev for editing. The results are shown in Table 4. It can be considered that after rephrasing, the ambiguous components in the instruction were minimized. Our method still demonstrates a clear improvement in consistency, which confirms our belief that for the IV-Complex task, fine-grained region guidance plays an important role in leveraging global semantics. The rephrase prompt we used is as follows:

> You are an expert AI assistant that rephrases complex image editing instructions into simple, direct commands.
> Your task is to convert the user's request into one or more concise and unambiguous commands that an image editing tool can understand.
> Follow these rules:
> 1. Directly extract the core action, the target object, and any specific attributes.
> 2. If the request involves multiple distinct steps, break it down into separate commands, one per line.
> 3. Keep the commands as short and direct as possible.
> 4. Your response must contain ONLY the rephrased command(s). Do not add any explanations, apologies, or conversational text.
> Instruction:

## E   ATTENTION RULE DISCOVERY

When experimenting with different attention rules, we observed several interesting phenomena.

First, if we cut off the attention across different regions of an image, very clear boundaries appear at the region edges, and the global consistency is lost, as shown in Fig 9.

Second, if we decouple the attention between image tokens (corresponding to the original image) and noise latent tokens, such that noise latent tokens can only attend to image tokens from specific regions, the model can then generate new images with reference to objects in those designated regions of the original image, as shown in Fig 10.

Third, if a particular image region does not receive any attention from the text modality, that part of the image suffers from severe distortion and noise. We hypothesize that the text modality also plays a role in facilitating internal information exchange within the image, as shown in Fig 11.

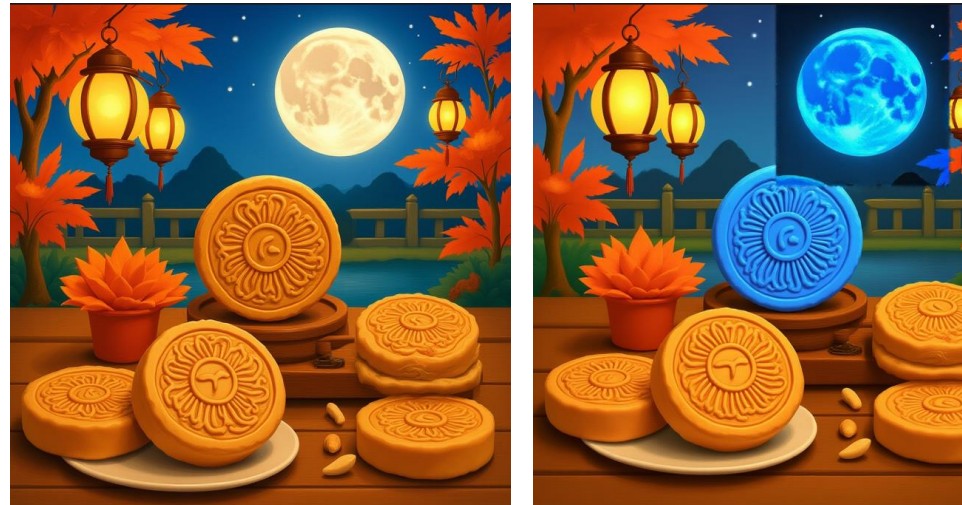

Figure 9: The right image is the original, and the left image shows the editing result where the attention between image patches of the editing region and the background is cut off. Clear regional boundaries can be observed.

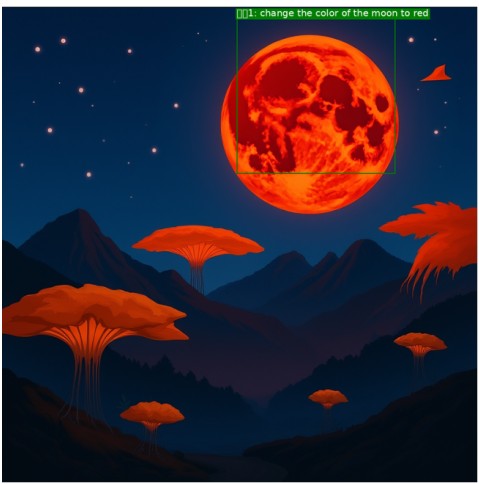

Figure 10: We decouple the noise patches and image patches, preserving their respective self-attention, but all image patches are only allowed to attend to the noise patches within the editing region. The resulting edited image, as shown in the figure, is able to retain the content and style of the original region.

## F    B-BOXES OVERLAPPING CASE

For cases where region bounding boxes overlap, our attention injection mechanism can also handle them correctly, as shown in the Figure 12, 13. The image patches within the overlapping areas can simultaneously attend to the corresponding hint text tokens. Moreover, since we preserve the full self-attention among all image patches, the model can autonomously manage the interactions between these sub-edits.

## G    ROBUSTNESS AGAINST B-BOX PERTURBATION

Since the success of our attention injection mechanism depends primarily on whether the b-box correctly corresponds to the target instance or region, pixel-level errors have little impact on the results. We conducted a perturbation experiment on the bboxes generated by the VLM, applying

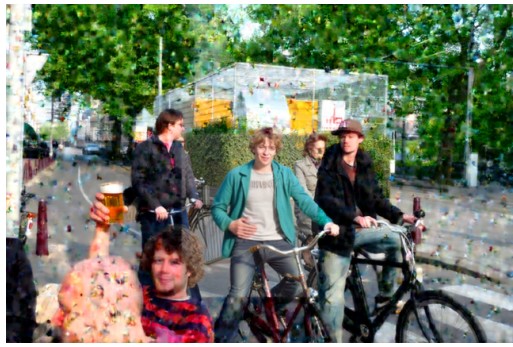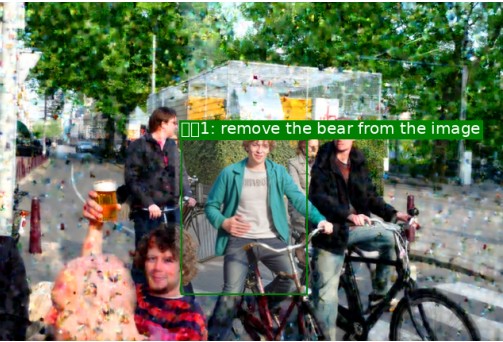

Figure 11: The left image shows the editing result, and the right image visualizes the editing region on top of the result. We mask out the background patches' attention to the global prompt token, and find that the image patches unable to attend to any text tokens exhibit severe distortion.

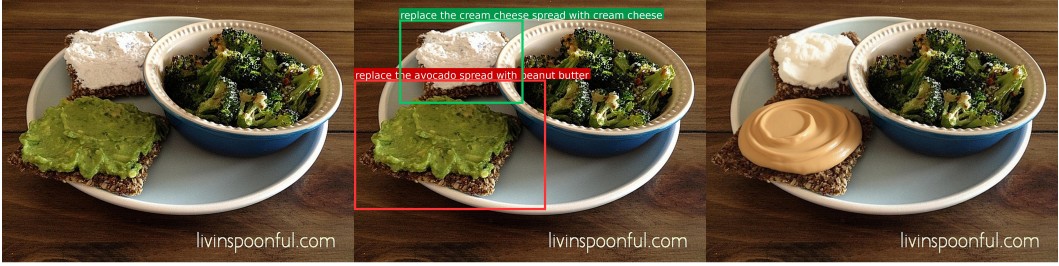

**Instruction:**

Replace the spread on the left cracker with peanut butter and the spread on the right cracker with cream cheese.

**VLM Response:**

<think>The user's instruction is to replace the spreads on the crackers. The left cracker has avocado spread, and the right cracker has cream cheese spread. The instruction specifies that the avocado spread should be replaced with peanut butter, and the cream cheese spread should be replaced with cream cheese.</think><gen_image>keep remaining part of image unchanged.</gen_image><region>[{"bbox_2d": [30, 175, 308, 358], "hint": "replace the avocado spread with peanut butter"}, {"bbox_2d": [108, 56, 282, 175], "hint": "replace the cream cheese spread with cream cheese"}]</region>

Figure 12: Overlapping Case 1

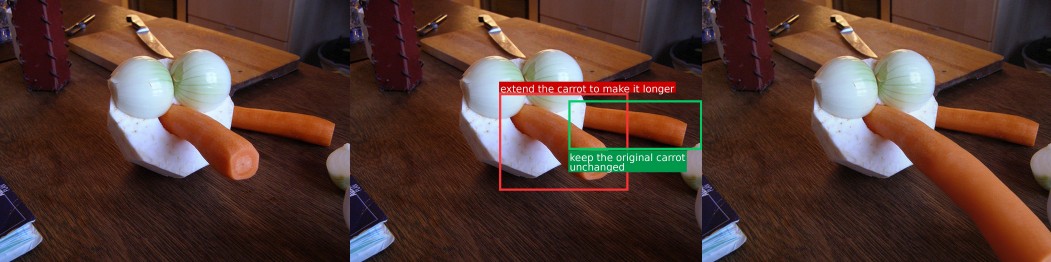

**Instruction:**

Imagine this vegetable creature is Pinocchio and it has just told a lie. Show what happens to the carrot acting as its nose.

**VLM Response:**

<think>The user wants the carrot to grow longer as a result of Pinocchio's lie, which is a classic fairy tale scenario.</think><gen_image>Keep remaining part of the image unchanged.</gen_image><region>[{"bbox_2d": [298, 189, 479, 326], "hint": "extend the carrot to make it longer"}, {"bbox_2d": [425, 192, 618, 262], "hint": "keep the original carrot unchanged"}]</region>

Figure 13: Overlapping Case 2

random scale and shift noise to all bbox corner points (for example, a 10% perturbation means each corner is shifted in a random direction by 10% of the bbox's width or height in pixels). The results are shown in table 5

Table 5: Results of b-box perturbation on VLM output of RePlan. This experiment is conducted using Flux.1 Kontext dev as MMDiT backbone.

| Perturbation Ratio | 0% | 10% | 20% | 50% | 70% |
|---|---|---|---|---|---|
| Overall | 3.46 | 3.46 | 3.45 | 3.45 | 3.35 |
| Weighted | 2.55 | 2.56 | 2.57 | 2.53 | 2.37 |

It can be seen that even when the perturbation ratio increases to 50%, our method still exhibits robustness.

# H MORE COMPARATIVE RESULTS

We provide additional comparative results in the Figure 14- 16 for reference.

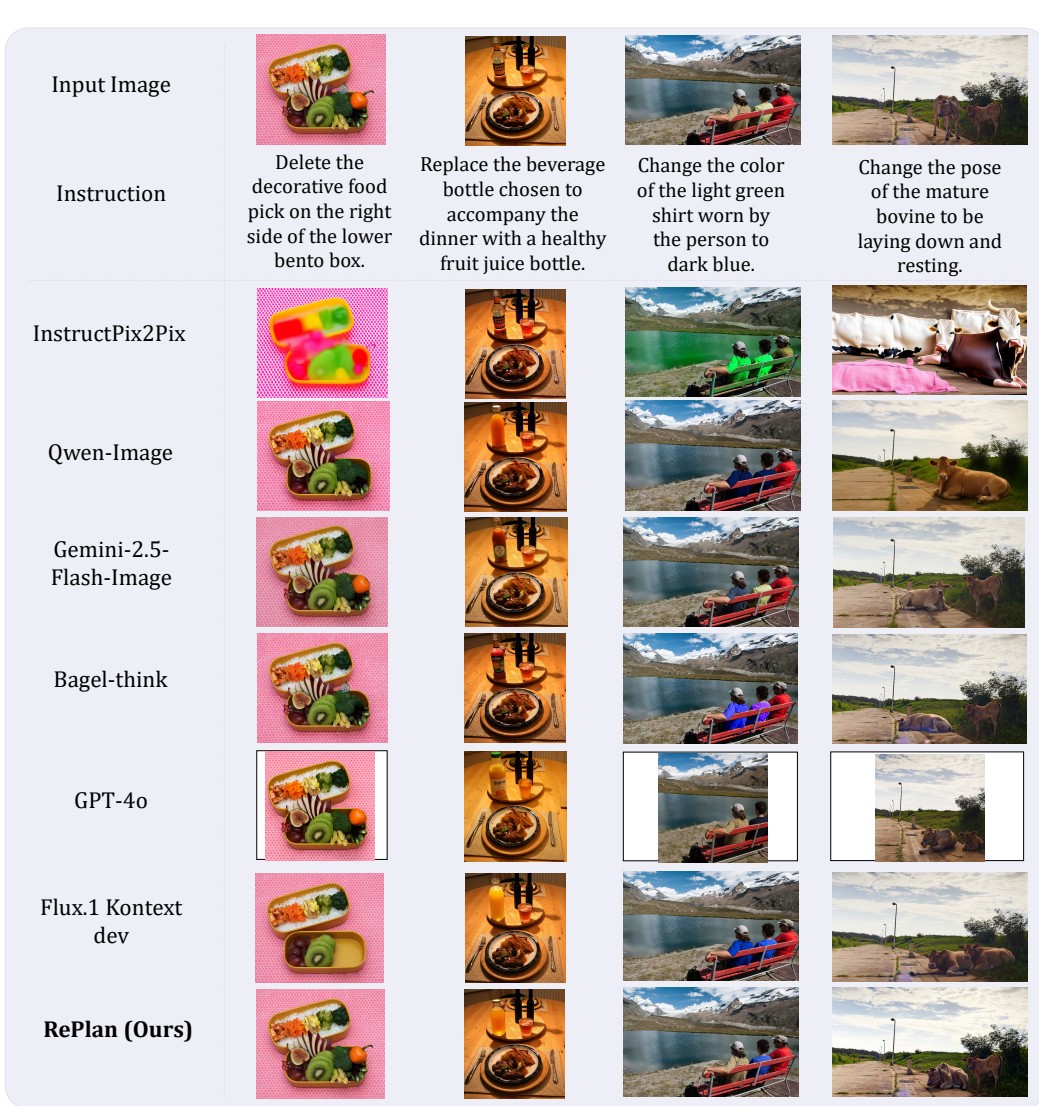

Figure 14: More Comparative Results. For the second column, the result of Flux.1 Kontext dev has a slight perspective change.

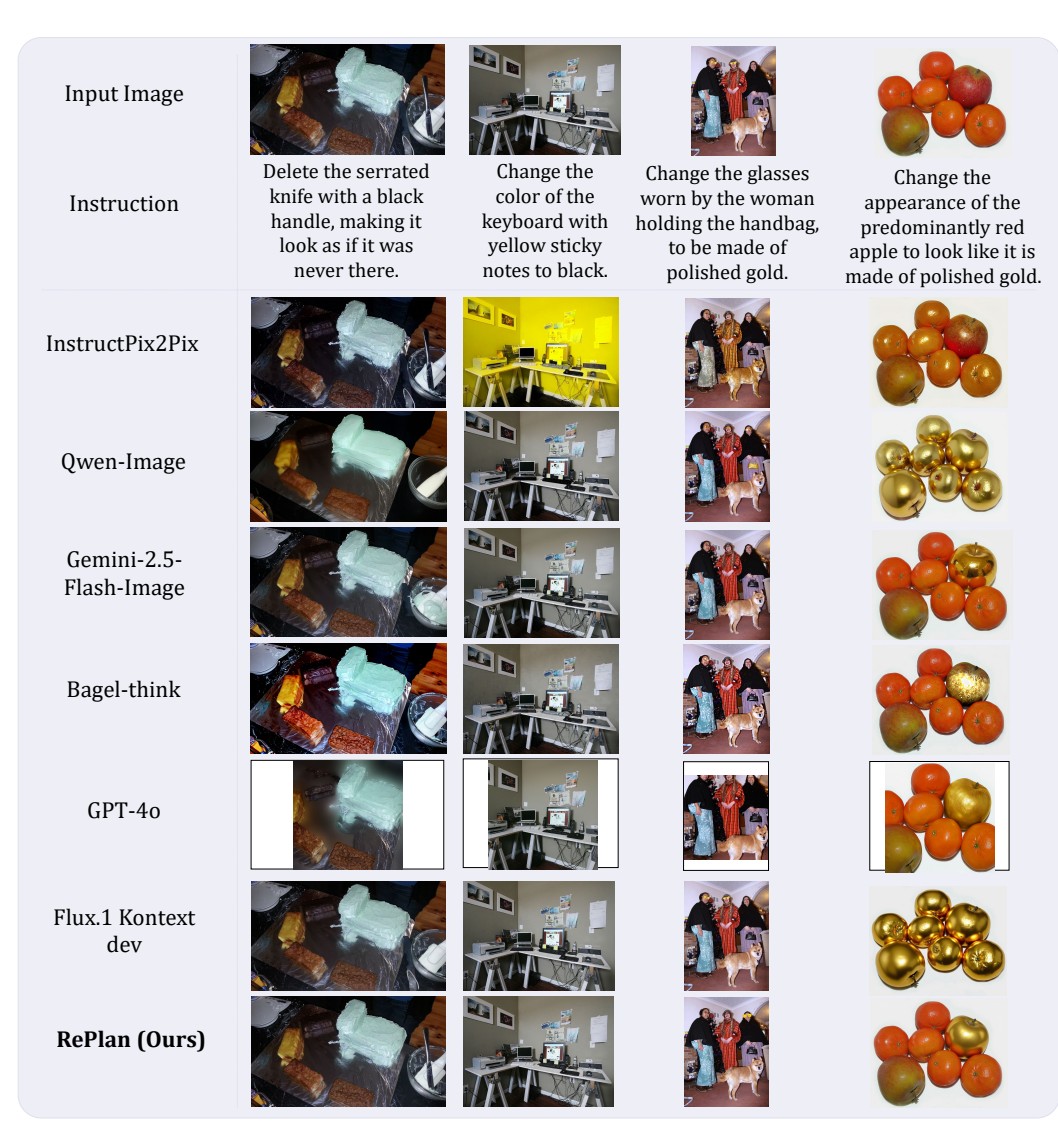

Figure 15: More Comparative Results.

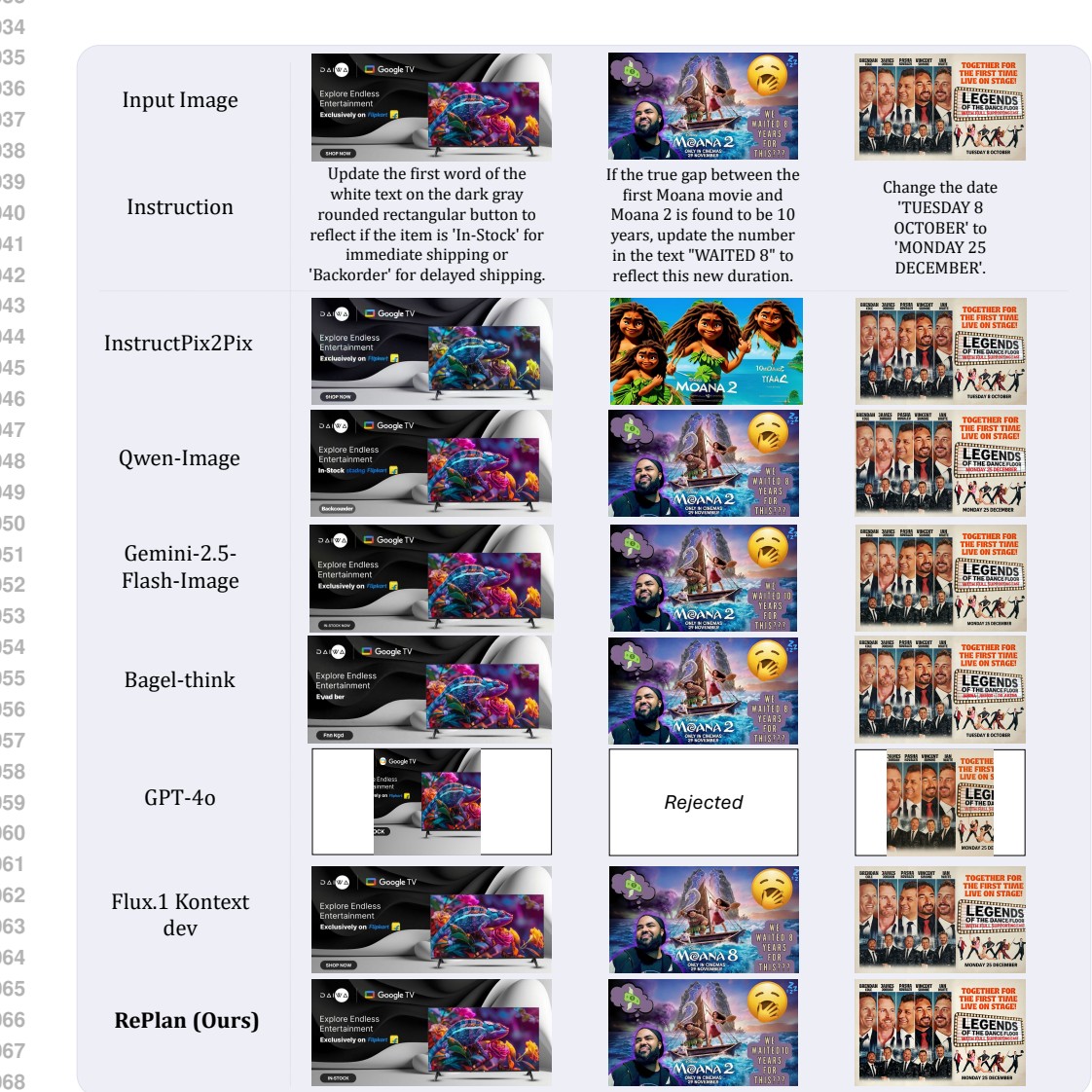

Figure 16: More Comparative Results under text editing scenario.

