# OpenReview forum: "RePlan: Reasoning-Guided Region Planning for Complex Instruction-Based Image Editing"
_ICLR.cc/2026/Conference — Submitted to ICLR 2026_

### Official Review · Reviewer_LjuJ · 2025-10-28

**Soundness:** 4
**Presentation:** 3
**Contribution:** 3
**Rating:** 4
**Confidence:** 3

**Summary:**

The paper proposes RePlan, a plan-then-execute framework for complex instruction-based image editing under Instruction–Visual Complexity. A VLM planner performs chain-of-thought reasoning to output region-aligned guidance and evaluate models with a VLM-as-judge on four dimensions, reporting that RePlan improves consistency and overall scores among open models; RL with GRPO on ~1k instruction-only samples further boosts planning reliability.

**Strengths:**

- Problem framing: IV-Complexity is clearly defined and motivated: The paper formalizes how visual clutter + instruction intricacy interact and argues why fine-grained grounding is essential—setting up the need for region-aware planning.
- Region-aligned planner with interpretable output & interactivity: The planner emits structured text with explicit tags and JSON region guidance, and authors highlight that users can adjust regions/hints—improving controllability and reproducibility of edits.
- Compact but effective RL for planning: A two-stage GRPO scheme first secures valid format/reasoning, then adds image-level rewards. Using ~1k instruction-only samples is attractive for data efficiency.

**Weaknesses:**

- Planner–evaluator entanglement in ablations: The planner ablation compares Gemini2.5-Pro and Qwen2.5-VL (7B) as zero-shot planners while evaluation still uses Gemini-2.5-Pro as judge. This can disadvantage the Qwen-planner setting (mismatched style) and—even for Gemini-planner—induces a same-family bias. Consider swapping judges to check rank stability.
- Limited dataset scale & unclear coverage of long-tail/sensitive cases: IV-Edit has ~800 pairs and emphasizes complexity, but the paper provides only high-level distributions. It would help to report fine-grained category/size/occlusion breakdowns, text font/layout diversity, and any sensitivity filtering procedures. A data card with bias/fairness diagnostics would increase credibility.
- Bounding-box robustness & error propagation not quantified: The approach hinges on bbox accuracy (planner → attention grouping). Authors note bbox errors (Gemini planner) qualitatively; however, there’s no systematic robustness test to bbox jitter, IoU drops, or mis-localized regions (impact on Target/Consistency). Please add stress tests that perturb boxes and report degradation curves.
- Latency/compute and scalability not reported: The method introduces planning (VLM) + editing (DiT with masked attention). Table 1 reports quality metrics only; no wall-clock comparisons across methods, nor scaling with number of regions (K). Reporting per-image latency and K-scaling (vs. iterative inpainting) would clarify practical trade-offs.

**Questions:**

N/A

---

> ### Author Response · Authors · 2025-11-22
> **(1/1) Thank you for the time and effort you dedicated to reviewing our work.**
>
> Thank you for the time and effort you dedicated to reviewing our work, below is our response for your concerns.
>
> ---
>
> **\[Q1\] Evaluator Bias of Planner Ablation**
>
> We additionally ran this ablation using GPT‑4o as the evaluator. The ranking remained stable. However, we observed that GPT‑4o has weaker fine‑grained perception ability compared with Gemini, which leads to overall inflated scores.
>
> | **Model**     | **quality** | **target** | **effect** | **consistency** | **overall** | **weighted** |
> | ------------- | ----------- | ---------- | ---------- | --------------- | ----------- | ------------ |
> | Qwen2.5-VL 7B | 3.38        | 2.40       | 1.81       | 3.32            | 2.73        | 1.81         |
> | Gemini2.5-pro | 3.86        | 2.97       | 2.31       | 3.82            | 3.24        | 2.25         |
> | Replan        | 3.91        | 3.48       | 2.90       | 4.24            | 3.63        | 2.82         |
>
> ---
> **\[Q2\] Dataset Scale & Category Distribution**
>
> **Most existing editing benchmarks are of a similar scale (for example, magicbrush: 1053, smartedit: 219).** We believe that 800 samples are sufficient to reasonably reflect an editing model’s fine‑grained complex editing capabilities as well as its world‑knowledge reasoning ability, while avoiding excessively high evaluation cost. We plan to expand to more domains in future work.
>
> Regarding the fine‑grained editing categories, we have already presented the numbers in Figure 5 (a) and (b). In addition, we will update these statistical results in the appendix of the revised PDF. The statistics for text layout and font are shown in the following table.
>
> Text layout：
>
> |           | **Tables** | **Slides** | **Photos** | **Posters** | **Drawings** | **Other** |
> | --------- | ---------- | ---------- | ---------- | ----------- | ------------ | --------- |
> | **Count** | 93         | 21         | 58         | 71          | 2            | 22        |
>
> Text font：
>
> |           | **Printed Font** | **Artistic Font** | **Handwriting** | **Other** | **Total** |
> | --------- | ---------------- | ----------------- | --------------- | --------- | --------- |
> | **Count** | 224              | 35                | 6               | 2         | 267       |
>
> The data source has already undergone sensitivity filtering for the images, and during our manual screening process we further applied additional sensitivity filtering.
>
> ---
> **\[Q3\] B-box Robustness**
>
> Since the success of our attention‑injection mechanism depends primarily on whether the bbox correctly corresponds to the target instance or region, pixel‑level deviations have limited impact on the outcome. We conducted perturbation experiments on the VLM‑generated bboxes by randomly scaling and shifting all bbox corners. A perturbation ratio of 10% means that each corner is displaced in a random direction by a distance equal to 10% of the bbox’s height or width. The results are as follows:
>
> |**Perturbation Ratio**|**0%**|**10%**|**20%**|**50%**|**70%**|
> |---|---|---|---|---|---|
> |Overall|3.46|3.46|3.45|3.45|3.35|
> |Weighted|2.55|2.56|2.57|2.53|2.37|
>
> As shown above, **even with perturbations as large as 50%, our method exhibits no significant performance degradation, demonstrating strong robustness to bbox inaccuracies.**
>
>
> ---
> **\[Q4\] Latency Comparison**
>
> We report the average inference latency of Replan and Kontext on IV-Edit, summarized in the table below：
>
> |**regions**|**1**|**2**|**3**|**4**|**5**|
> |---|---|---|---|---|---|
> |Replan|47.0s|47.6s|48.5s|49.2s|50.0s|
> |multi-turn flux kontext|44.2s|88.4s|132.6s|176.8s|221.0s|
>
> For Replan, the increase in latency with larger k mainly comes from the growing number of text tokens processed by the VLM and MMDiT. In contrast, under the multi‑turn setting, each turn requires forwarding all image patches, leading to a roughly linear growth in time cost as k increases. **This reflects the computational efficiency of our method.**

---

> ### Author Response · Authors · 2025-11-25
> **Looking forward to you feedback!**
>
> Dear Reviewer LjuJ,
>
> We truly appreciate the time and care you devoted to reviewing our work. As the discussion deadline approaches, we would be grateful if you could let us know whether our responses sufficiently address your concerns, or if you have any additional thoughts.
>
> Sincerely,
> The Authors

---

### Official Review · Reviewer_SvSJ · 2025-11-01

**Soundness:** 3
**Presentation:** 3
**Contribution:** 3
**Rating:** 6
**Confidence:** 3

**Summary:**

This paper presents RePlan, a reasoning-guided image editing framework designed to handle Instruction–Visual Complexity (IV-Complexity)—where complex textual instructions must be grounded in visually cluttered scenes. The framework couples a vision–language model (VLM) planner and a diffusion-based decoder, using a training-free attention region injection mechanism for precise, parallel, multi-region edits. Furthermore, the paper introduces IV-Edit, a new benchmark for evaluating complex instruction-based editing. Experiments show that RePlan outperforms several state-of-the-art baselines in consistency and reasoning-based grounding.

**Strengths:**

1. The introduction of “Instruction–Visual Complexity” formalizes an important challenge in multimodal editing and provides clear motivation for a new benchmark.

2. The combination of reasoning-driven planning with region-level attention control is well thought out and technically sound, avoiding costly retraining.

3. The new IV-Edit benchmark and extensive comparisons with both open-source and proprietary models strengthen the empirical validation.

**Weaknesses:**

1. Unclear realization of “step-by-step reasoning”: Although the paper claims that the planner decomposes instructions via step-by-step reasoning, neither the framework diagram nor the visualization results  clearly illustrate this multi-step reasoning process. The reasoning seems to occur implicitly rather than explicitly, weakening the claimed interpretability advantage.

2. The IV-Edit benchmark (∼800 samples) may not be large enough to comprehensively assess generalization, especially for reasoning-intensive tasks.

3. While RePlan improves consistency, the overall quantitative gains over strong baselines are modest, and some reinforcement learning details (e.g., reward weights, training stability) are insufficiently documented for reproducibility.

**Questions:**

Please see the weaknesses.

---

> ### Author Response · Authors · 2025-11-22
> **(1/1)Thank you for your thoughtful suggestions and valuable feedback**
>
> Thank you for your thoughtful suggestions and valuable feedback, below is our response for your concerns.
>
> ---
>
> **\[Q1\] ‘Step-by-step’ Clarification**
>
> Our intended meaning behind “step-by-step reasoning” is explicitly as follows:
> 1. The VLM first understands the user instruction and image content (implicit).
> 2. It then analyzes the true intent of the user instruction (implicit).
> 3. It generates planning results and region–hint pairing.
> 4. It executes the final edits.
> This process corresponds to the upper part of Figure 2. We will also revise Figure 2 slightly to make this multi-step reasoning flow more intuitive.
>
> ---
> **\[Q2\] Data Scale**
>
> **Existing editing benchmarks are generally of a similar scale (e.g., MagicBrush: 1053 samples, SmartEdit: 219 samples).** We believe that a set of 800 samples is sufficient to reasonably reflect a model’s fine-grained editing capability and its world-knowledge reasoning ability, **while avoiding excessive evaluation cost**. We plan to extend the benchmark with additional domains and a larger number of samples to further improve the assessment of generalization in the future.
>
> ---
> **\[Q3\] Gains Over Strong Baselines & Training Details**
>
> We replaced Flux Kontext Dev in RePlan with Qwen-Image for additional experiments. The results show that our method yields clear improvements across different MMDiT baselines:
>
> | **Model**                 | **Quality ↑** | **Target ↑** | **Effect ↑** | **Consistency ↑** | **Overall ↑** | **Weighted ↑** |
> | ------------------------- | ------------- | ------------ | ------------ | ----------------- | ------------- | -------------- |
> | **Flux.1 Kontext Dev**    | 3.93          | 3.34         | 2.73         | 2.88              | 3.22          | 2.49           |
> | **RePlan (Flux Kontext)** | 4.16          | 3.47         | 2.59         | 3.64              | 3.46          | 2.55           |
> | **Qwen-Image**            | 3.47          | 3.72         | 3.24         | 1.79              | 3.05          | 2.62           |
> | **RePlan (Qwen-Image)**   | 3.86          | 3.77         | 3.16         | 3.24              | 3.51          | 2.91           |
>
> Regarding reproducibility, we will release the full training and evaluation code (including all reward weights and training parameters) as well as the training data.

---

> ### Author Response · Authors · 2025-11-25
> **Looking forward to your feedback!**
>
> Dear Reviewer SvSJ,
>
> Thank you very much for your valuable comments. We would greatly appreciate hearing whether our revisions address your concerns, or whether you have any further suggestions before the discussion deadline.
>
> Best regards,
> The Authors

---

### Official Review · Reviewer_Tvuy · 2025-11-08

**Soundness:** 2
**Presentation:** 2
**Contribution:** 3
**Rating:** 4
**Confidence:** 4

**Summary:**

This paper formalizes the challenge of Instruction-Visual Complexity and addresses it from a unified model perspective. Unlike existing unified models such as Bagel, the authors propose a region-level approach. They also introduce a training-free mechanism called Attention Region Injection and establish a new benchmark to evaluate this problem.

**Strengths:**

1. The proposed region-level control is both necessary and interesting.

2. A new benchmark is introduced to evaluate the proposed task.

3. Qualitative results demonstrate clear improvements over previous methods.

**Weaknesses:**

1. Figures 9–12 are confusing, as they are not referenced in the text nor sufficiently explained in the captions.

2. I wonder whether using the generated text from the VLM as input to existing MMDiT-style editing models (e.g., Flux-Edit) would still improve performance. This raises the question of whether the effectiveness comes from the comprehensive textual information or from the training-free editing mechanism itself.

3. In Stage 1 (RL), it is unclear how the rewards are computed and what the tag format looks like.

4. More conventional editing metrics or benchmarks should be included. Relying solely on the newly proposed benchmark makes the evaluation less convincing. I assume the proposed method should also perform well on existing editing benchmarks.

5. Qwen-Image shows better weighted results, which calls into question the advantages of the proposed method, given that Qwen-Image also adopts an MMDiT-based unified architecture.

6. Beyond Figures 1 and 6, more comparative results should be presented. Additionally, I could not find any supplementary results.

**Questions:**

See Weaknesses

---

> ### Author Response · Authors · 2025-11-22
> **(1/3) Thank you for the time you invested in the review and for raising very pertinent questions.**
>
> Thank you for the time you invested in the review and for raising very pertinent questions, below is our response for your concerns.
>
> ---
> **\[W1\] Captions Revision**
>
> Thank you for pointing this out. We have corrected the captions for Figure 9–12 in the revised PDF.
>
> ---
> **\[W2\] Textual Information & Attention Injection Ablation**
>
> To our best knowledge, we have not been able to find "Flux-Edit" or any related paper mentioning it. Could you kindly provide the relevant references? We will first conduct experiments using Flux Kontext dev with generated text as input. In our experiments, **we use the same text input sequence as replan, but without applying attention injection**. The results are as follows:
>
> |                         | quality | target | effect | consistncy | overall | weighted |
> | :---------------------- | :------ | :----- | :----- | :--------- | :------ | :------- |
> | replan                  | 4.16    | 3.47   | 2.59   | 3.64       | 3.46    | 2.55     |
> | w/o attention injection | 3.81    | 3.09   | 2.34   | 2.89       | 3.03    | 2.17     |
>
> **These results show that our attention injection mechanism plays a crucial role. **
>
> **Existing MMDiT-style models cannot accept our region-aligned guidance without our training-free mechanism.** Our guidance consists of b-boxes and their corresponding region hint pairs. If these are directly concatenated as text without referring expressions, the diffusion model has no way to determine which instruction corresponds to which editing region.
> **In addition, these MMDiT models that use CLIP as the text encoder can only process at most 77 tokens.** If VLM-generated full text is directly fed as input without disentangling region-level instructions, the excess text will be truncated.
>
> ---
> **\[W3\] RL Reward Computation**
>
> Tag format can refer to Figure 3.
> - Tag format reward: We use regular expression matching to determine whether the output is organized in the structured format shown in Figure 3, including all “\<tag\>”. If parsing succeeds, the reward is set to 1; otherwise it is 0.
> - Region format reward: We use JSON parsing on the text inside the "\<region\>" tag, including the outer list and the inner dicts (each dict contains two items: bbox and hint). If parsing fails, the reward is 0; otherwise it is 1.
> - Reasoning quality reward: The think part receives a reward proportional to its length. When the length reaches 50, it achieves the maximum value of 0.25.
>
> ---

---

> ### Author Response · Authors · 2025-11-22
> **(2/3)**
>
> **\[W4\] Extra Metrics**
>
> **Extra Metric:** We recruited 12 human experts to conduct the user study. From IV‑edit, we randomly sampled 100 instruction–image pairs and compared the results of flux kontext dev and replan. During the entire process, the human experts were blind to which model produced which result. We asked the experts to vote on which of the two edited results better followed the editing instruction, or indicate that the two were comparable. For each instruction–image pair, the option with the most votes was taken as the label. We obtained the following statistics:
>
> | replan better | comparable | kontext better |
> | ------------- | ---------- | -------------- |
> | 35%           | 50%        | 15%            |
>
> **The user‑study results, provided as an additional metric, also demonstrate the effectiveness of our method.**
>
> **Other Benchmarks:** **We constructed our benchmark because existing instruction‑editing benchmarks do not effectively evaluate how editing models handle IV‑Complex tasks.** To further demonstrate that our method maintains generalization ability on other existing benchmarks, we additionally conducted experiments on MagicBrush and SmartEdit (Reason‑Edit).
>
> MagicBrush：
>
> | **Methods**                        | **L1$\downarrow$** | **L2$\downarrow$** | **CLIP-I$\uparrow$** | **DINO$\uparrow$** | **CLIP-T$\uparrow$** |
> | ---------------------------------- | ------------------ | ------------------ | -------------------- | ------------------ | -------------------- |
> | HIVE                               | 0.1092             | 0.0341             | 0.8519               | 0.7500             | 0.2752               |
> | HIVE w/ MagicBrush                 | 0.0658             | 0.0224             | 0.9189               | 0.8655             | 0.2812               |
> | InstructPix2Pix                    | 0.1122             | 0.0371             | 0.8524               | 0.7428             | 0.2764               |
> | InstructPix2Pix w/ MagicBrush      | 0.0625             | 0.0203             | 0.9332               | _0.8987_           | 0.2781               |
> | **Kontext**                        | 0.0654             | 0.0259             | 0.9193               | 0.8608             | **0.2913**           |
> | **Kontext w/ attention injection** | **0.0516**         | **0.0188**         | **0.9356**           | 0.8978             | _0.2878_             |
> | **Replan**                         | _0.0539_           | _0.0197_           | _0.9337_             | **0.9014**         | 0.2866               |
>
> SmartEdit (reasoning)：
>
> | Methods                            | PSNR(dB)↑  |   SSIM↑   |  LPIPS↓   | CLIP Score↑ | Ins-align↑ |
> | :--------------------------------- | :--------: | :-------: | :-------: | :---------: | :--------: |
> | InstructPix2Pix                    |   24.234   |   0.707   |   0.083   |   19.413    |   0.344    |
> | MagicBrush                         |   22.101   |   0.694   |   0.113   |   19.755    |   0.283    |
> | InstructDiffusion                  |   21.453   |   0.666   |   0.117   |   19.523    |   0.483    |
> | SmartEdit-7B                       |   25.258   |   0.742   |   0.055   | **20.950**  |   0.789    |
> | **Kontext**                        |   23.764   |   0.779   |   0.050   |   20.251    |   0.817    |
> | **Kontext w/ attention injection** |  _30.82_   |  _0.964_  |  _0.021_  |   20.662    | **0.933**  |
> | **Replan**                         | **31.533** | **0.967** | **0.019** |  _20.769_   |  _0.900_   |
>
> In the two tables, `“Kontext w/ attention injection”` refers to the results obtained by using the human‑annotated masks provided by the benchmark as region guidance and applying attention injection. **The results on both benchmarks show that replan and attention injection yield clear improvements over Kontext.**

---

> ### Author Response · Authors · 2025-11-22
> **(3/3)**
>
> **\[W5\] Advantages of Our Proposed Method**
>
> We believe that even if the weighted score of Qwen-Image is higher than that of Replan, **this does not diminish the unique advantages of our method. The final performance of our approach is influenced by the choice of the MMDiT backbone.** Qwen-Image uses a 20B MMDiT, which is significantly larger than the 12B MMDiT in Flux Kontext dev. Results from IV-edit and other existing benchmarks consistently show that Qwen-Image outperforms Kontext dev, and this observation is also widely recognized by the user community.
>
> We **replaced** `Flux Kontext dev` with `Qwen-Image` within Replan and conducted experiments. The results show that our method achieves clear improvements across different MMDiT backbones, which **sufficiently demonstrates the advantages and transferability of our approach**：
>
> | **Model**                 | **Quality ↑** | **Target ↑** | **Effect ↑** | **Consistency ↑** | **Overall ↑** | **Weighted ↑** |
> | ------------------------- | ------------- | ------------ | ------------ | ----------------- | ------------- | -------------- |
> | **Flux.1 Kontext Dev**    | 3.93          | 3.34         | 2.73         | 2.88              | 3.22          | 2.49           |
> | **RePlan (Flux Kontext)** | 4.16          | 3.47         | 2.59         | 3.64              | 3.46          | 2.55           |
> | **Qwen-Image**            | 3.47          | 3.72         | 3.24         | 1.79              | 3.05          | 2.62           |
> | **RePlan (Qwen-Image)**   | 3.86          | 3.77         | 3.16         | 3.24              | 3.51          | 2.91           |
>
> ---
> **\[W6\] More Comparative Results**
>
> Thank you for your suggestion. We have added more comparative visual results in appendix H of the revised PDF.

---

> ### Author Response · Authors · 2025-11-25
> **Looking forward to your feedback!**
>
> Dear Reviewer Tvuy,
>
> We sincerely appreciate your thoughtful suggestions and have carefully revised our responses. As the discussion‑phase deadline nears, we would be thankful if you could let us know whether our clarifications resolve your concerns or if you have any additional feedback.
>
> Respectfully,
> The Authors

---

### Official Review · Reviewer_YTWG · 2025-11-09

**Soundness:** 3
**Presentation:** 3
**Contribution:** 3
**Rating:** 6
**Confidence:** 4

**Summary:**

This paper addresses the challenge of Instruction-Visual Complexity (IV-Complexity), which arises from the interplay between visual complexity (e.g., cluttered layouts, multiple similar objects) and instructional complexity (e.g., multi-object references, implicit semantics, world knowledge, and causal reasoning requirements) in image editing tasks. The authors propose refining the interaction between Vision-Language Models (VLMs) and diffusion models from a global semantic level to a region-specific level, leveraging VLMs' fine-grained perception and reasoning capabilities to generate region-aligned guidance.
Subsequently, the authors introduce RePlan, a framework that adopts a two-stage plan-execute paradigm: the VLM performs chain-of-thought reasoning to analyze the input image and instruction, producing structured region-aligned guidance (comprising bounding boxes and editing prompts); the diffusion model then executes precise multi-region parallel editing through a training-free attention-based region injection mechanism, while GRPO reinforcement learning (only ~1k instruction-only samples) enhances the VLM's planning capability. Experimental results demonstrate that RePlan outperforms existing models trained on massive-scale data on the newly proposed IV-Edit benchmark, achieving state-of-the-art performance among open-source models, effectively mitigating the issue of edit leakage to similar regions.

**Strengths:**

# Originality
- The paper systematically introduces the concept of "Instruction-Visual Complexity (IV-Complexity)," explicitly defining the challenges arising from the interplay between visual complexity (cluttered layouts, multiple similar objects) and instructional complexity (multi-object references, implicit semantics, knowledge reasoning).
- Through attention mask rules (prompt isolation, region constraints, background constraints, etc.), the method achieves precise multi-region parallel editing without requiring retraining of the DiT model.
- The construction of the IV-Edit benchmark represents the first evaluation dataset
# Quality
- The paper compares 2 closed-source models (GPT-4o, Gemini) and 6 open-source models on the IV-Edit benchmark, employing Gemini 2.5-Pro as the evaluator with 5-point scoring across four dimensions: target localization, consistency, quality, and effect.
- Ablation studies validate the necessity of both RL training and chain-of-thought (CoT) reasoning.
# Clarity
The problem is articulated clearly, the methodology is described in reasonable detail, and there are no apparent grammatical errors.
# Significance
- The problem formulation is well-defined, providing novel methods and benchmarks for complex instruction-based editing.
- The interpretability and interactivity of region-guided editing enhance the practical utility of the system.

**Weaknesses:**

- All main results (Table 1) rely on 5-point scoring from a single closed-source model, presenting risks that Gemini may exhibit bias towards its own model (Gemini-Flash-Image) with more lenient scoring, or that results may not be reproducible.

- The paper does not specify the image sources, annotation procedures, or other critical aspects of the dataset construction process.

- Key hyperparameters such as learning rate, batch size, number of training epochs, and GRPO group size are not documented in either the main text or the appendix.

**Questions:**

1. RePlan relies on VLM-generated bounding boxes (bboxes), yet the paper does not discuss the importance of bbox accuracy. It remains unclear whether employing existing detection models such as Grounding DINO or Grounding SAM to provide more precise region bounding boxes would yield comparable or superior performance.

2. When multiple target regions spatially overlap or are highly adjacent, what impact does this have on the editing results? The paper lacks analysis of how the attention mask mechanism handles boundary conflicts in such scenarios.

3. How is the VLM training dataset constructed? The paper does not provide sufficient details regarding data collection, annotation procedures, or quality control measures.

4. Does the attention-based region injection mechanism depend on MMDiT-specific designs (e.g., joint text-image attention)? Can this approach be transferred to other editing models with different architectural backbones, such as UNet-based diffusion models?

5. IV-Edit primarily comprises realistic photographs and document editing tasks. The benchmark could be further extended to encompass artistic styles (e.g., cartoons, oil paintings) or specialized professional domains such as medical imaging.

6. User studies could be incorporated as an objective evaluation metric to assess whether model behavior aligns with human preferences, thereby providing a more comprehensive assessment beyond automated metrics.

7. The VLM primarily replaces the manual annotation of bboxes and the input of localized editing prompts by humans. However, if human-annotated bboxes and prompts are directly provided, could Flux Kontext achieve precise localized editing without requiring any training data? It would be beneficial to include visualization or quantitative experiments as reference while explicitly articulating the unique advantages of employing a VLM in this pipeline.

---

> ### Author Response · Authors · 2025-11-22
> **(1/2) Thank you for your insightful and detailed comments.**
>
> Thank you for your insightful and detailed comments, below is our response for your concerns.
>
> ---
> **\[Q1\] Discussion on B-boxes Accuracy**
>
> **B-box accuracy:** Since the success of our attention injection mechanism depends primarily on whether the b-box correctly corresponds to the target instance or region, pixel-level errors have little impact on the results. We conducted a perturbation experiment on the bboxes generated by the VLM, applying random scale and shift noise to all bbox corner points (for example, a 10% perturbation means each corner is shifted in a random direction by 10% of the bbox’s width or height in pixels). The results are as follows:
>
> |**Perturbation Ratio**|**0%**|**10%**|**20%**|**50%**|**70%**|
> |---|---|---|---|---|---|
> |Overall|3.46|3.46|3.45|3.45|3.35|
> |Weighted|2.55|2.56|2.57|2.53|2.37|
>
> **It can be seen that even when the perturbation ratio increases to 50%, our method still exhibit robustness.**
>
> **VLM instead of detection models:** The core reason we choose to use a VLM to generate b-boxes is that we need to leverage its strong language understanding, world knowledge, and reasoning abilities to interpret user editing instructions and plan editing hints aligned with specific regions. **Models like Grounding DINO and Grounding SAM struggle to achieve this.** In addition, Qwen2.5-VL has already achieved performance comparable to detection-specialized models like Grounding DINO on traditional detection/grounding tasks, as shown below:
>
> |Dataset|Refcoco val|Refcoco testA|Refcoco testB|Refcoco+ val|Refcoco+ testA|Refcoco+ testB|Refcocog val|Refcocog test|
> |:--|:-:|:-:|:-:|:-:|:-:|:-:|:-:|:-:|
> |**Grounding DINO**|90.6|93.2|88.2|88.2|89.0|75.9|86.1|87.0|
> |**Qwen2.5-VL 7B**|90.0|92.5|85.4|84.2|89.1|76.9|87.2|87.2|
>
> In summary, we believe that introducing VLMs to generate b-boxes is necessary.
>
> ---
> **\[Q2\] Overlapping Case**
>
> We have added **visualization results** in the appendix section F of the revised PDF, showing that our method can still perform reliable parallel multi‑region editing even when the target regions overlap.
> When overlaps occur, the overlapping regions’ image patches can simultaneously receive attention from multiple region hints. Since we preserve the full self-attention among all image patches, **this introduces no additional impact**.
> (Imagine a standard instruction-editing model performing the task “turn the person’s hat red and make their clothes white” _without_ using our attention injection method. This is equivalent to, in our approach, using two bounding boxes the size of the full image and completely overlapping, each corresponding to the first and second parts of the instruction respectively. )
>
>
> ---
> **\[Q3\] Data Annotation Pipeline**
>
> We have updated the detailed data construction pipeline in Appendix Section C of the revised PDF.
>
> ---
> **\[Q4\]  Transferring to Other MMDiT Backbone**
>
> Yes. Regarding the joint text-image attention design in MMDiT: we replaced Flux Kontext in our method with Qwen-image-edit, which is also based on MMDiT. The experiments show that our method achieves similarly **significant improvements**:
>
> | **Model**                 | **Quality ↑** | **Target ↑** | **Effect ↑** | **Consistency ↑** | **Overall ↑** | **Weighted ↑** |
> | ------------------------- | ------------- | ------------ | ------------ | ----------------- | ------------- | -------------- |
> | **Flux.1 Kontext Dev**    | 3.93          | 3.34         | 2.73         | 2.88              | 3.22          | 2.49           |
> | **RePlan (Flux Kontext)** | 4.16          | 3.47         | 2.59         | 3.64              | 3.46          | 2.55           |
> | **Qwen-Image**            | 3.47          | 3.72         | 3.24         | 1.79              | 3.05          | 2.62           |
> | **RePlan (Qwen-Image)**   | 3.86          | 3.77         | 3.16         | 3.24              | 3.51          | 2.91           |
>
>
> ---
> **\[Q5\] Domain Extension**
>
> Thank you for your suggestion. Our data pipeline can be easily extended to other domains, and we plan to further expand the diversity in future work.
>
>
> ---
> **\[Q6\] User Study**
>
> We recruited 12 human experts to conduct the user study. From IV‑edit, we randomly sampled 100 instruction–image pairs and compared the results of flux kontext dev and replan. During the entire process, the human experts were blind to which model produced which result. We asked the experts to vote on which of the two edited results better followed the editing instruction, or indicate that the two were comparable. For each instruction–image pair, the option with the most votes was taken as the label. We obtained the following statistics:
>
> | Replan Better | Comparable | Kontext Better |
> | ------------- | ---------- | -------------- |
> | 35%           | 50%        | 15%            |
>
> These results provide comprehensive evidence of the advantage of our method.

---

> ### Author Response · Authors · 2025-11-22
> **(2/2)**
>
> **\[Q7\] Unique Advantages of Employing VLM**
>
> The advantage of introducing a VLM is that it can automatically process a large number of images using instruction templates without requiring human annotation, which is also the advantage of instruction-based image editing. Because only the instruction is needed as input, replan can be applied in various automated agentic systems, while **manually providing bboxes and region‑level prompts for every image is extremely time‑consuming**.
>
> Nevertheless, in task scenarios that require human involvement for fine‑grained editing, **our attention injection method still allows Flux Kontext to accept human‑annotated bboxes and prompts in a training‑free manner**. We conducted experiments on MagicBrush. In the table, `“kontext w/ attention injection”` refers to the results obtained by using the human‑annotated masks provided by MagicBrush as region guidance and applying attention injection.
>
> | **Methods**                    | **L1$\downarrow$** | **L2$\downarrow$** | **CLIP-I$\uparrow$** | **DINO$\uparrow$** | **CLIP-T$\uparrow$** |
> | ------------------------------ | ------------------ | ------------------ | -------------------- | ------------------ | -------------------- |
> | HIVE                           | 0.1092             | 0.0341             | 0.8519               | 0.7500             | 0.2752               |
> | HIVE w/ MagicBrush             | 0.0658             | 0.0224             | 0.9189               | 0.8655             | 0.2812               |
> | InstructPix2Pix                | 0.1122             | 0.0371             | 0.8524               | 0.7428             | 0.2764               |
> | InstructPix2Pix w/ MagicBrush  | 0.0625             | 0.0203             | 0.9332               | _0.8987_           | 0.2781               |
> | kontext                        | 0.0654             | 0.0259             | 0.9193               | 0.8608             | **0.2913**           |
> | kontext w/ Attention injection | **0.0516**         | **0.0188**         | **0.9356**           | 0.8978             | _0.2878_             |
> | replan                         | _0.0539_           | _0.0197_           | _0.9337_             | **0.9014**         | 0.2866               |
> **The results are comparable to Replan and substantially outperform Kontext.**

---

> ### Author Response · Authors · 2025-11-25
> **Looking forward to your feedback!**
>
> Dear Reviewer YTWG,
>
> Thank you again for your detailed and insightful comments. We would be very grateful if you could kindly let us know whether our responses have addressed your concerns, or if you have any further feedback as the discussion deadline approaches.
>
> Sincerely,
> The Authors

---

### Author Response · Authors · 2025-11-22
**Gloabal Response**

We sincerely thank all reviewers for the time and effort dedicated to evaluating our work and for the thoughtful questions and suggestions provided. Your feedback has been highly valuable in helping us clarify the presentation, strengthen the experimental analysis, and refine the overall contribution of the paper. We have carefully addressed each comment in the following sections and revised the manuscript accordingly.

---

### Author Response · Authors · 2025-12-01
**Rebuttal Summary for the New AC and Reviewers (3/3)**

### **Reviewer Tvuy:**

> **\[W1\] Unclear referencing and description of Figures 9–12**

In the revised manuscript, we **completed the in-text references** to Figures 9–12 and **rewrote the captions**.

> **\[W2\] Source of improvement: stronger text vs. training-free attention**

We conducted the ablation reviewer metioned: **without attention injection**, we fed the **same VLM-generated text** used in RePlan into the original `Flux Kontext`.

The scores remain clearly lower than with RePlan. This indicates that the main gain comes from our **attention injection**, not just longer or more detailed text.

> **\[W3\] Stage-1 RL reward design and label format**

This mainly concerns **label structure and reward implementation**. We have updated the description in the revised PDF and will **open-source all details in code** to ensure reproducibility.

> **\[W4\] More metrics and existing benchmarks**

Beyond IV-Edit, we added two kinds of validation:
- A **12-expert user study** to directly confirm that RePlan is preferred over Kontext,
- and **evaluation on `MagicBrush` and `SmartEdit`**.

Both the “`Kontext` + attention injection” setting with human masks and the full RePlan pipeline **significantly outperform the original `Flux Kontext`**, showing **strong cross-benchmark generalization**.

> **\[W5\] Higher Weighted score of `Qwen-Image` and doubts about our advantage**

- Applying RePlan to `Flux Kontext Dev` yields **clear improvements over the original Kontext**, which already demonstrates the method’s effectiveness.
- Since `Qwen-Image` and `Flux Kontext Dev` differ greatly in **base performance and parameter scale** , a slightly higher Weighted score for Qwen-Image is expected.
- Moreover, when we **plug-and-play RePlan on `Qwen-Image`**, its metrics further **improve significantly** over the original `Qwen-Image`, confirming that our **region-level guidance benefits different MMDiT backbones**.

> **\[W6\] More visual results and supplementary analysis**

As suggested, we **added more comparative visualizations** in the main text and **richer analyses in the supplementary material**.

---
### **Reviewer SvSJ:**

> **\[W1\] Ambiguity of “step-by-step reasoning”**

By `step-by-step`, we specifically mean **four stages**: (i) understanding the image and instruction, (ii) analyzing the true editing intent, (iii) planning region–hint pairs, and (iv) executing the edits. We **updated Fig. 2** to more clearly annotate this multi-step pipeline.

> **\[W2\] Is IV-Edit sufficient for assessing reasoning generalization?**

- We note that IV-Edit is **similar in size** to existing editing benchmarks.
- We argue that its samples are sufficient to distinguish current models on IV-complex tasks **without excessive evaluation cost**.
- We also support our generalization claim with **experiments on MagicBrush and SmartEdit**, where RePlan performs well on external benchmarks.

> **\[W3\] Limited quantitative gains over strong baselines and RL details**

(i) Quantitative gains:
- Our gains on **Consistency and overall scores** are **stable and clear** over the backbone `Flux.1 Kontext Dev`.
- The relatively smaller gains of `RePlan(Kontext)` compared to `Qwen-Image` are largely due to the **backbones’ performance gap**.
- After applying RePlan to `Qwen-Image`, we observe **even more pronounced improvements**.

(ii) RL parameter details:
- For reproducibility, we will **open-source the full training/evaluation code and data**, and we will add **parameter and training details** in the revised PDF.
---
### **Reviewer LjuJ:**

> **\[W1\] Potential evaluation bias in ablations Table 2**

We **replaced the evaluator with `GPT-4o`** and re-ran the evaluation. The **relative ranking remains stable** across settings.

> **\[W2\] Limited size of IV-Edit and need for more statistics**

- We emphasize that IV-Edit is **comparable in size** to mainstream editing benchmarks (as noted above).
- We also added **more fine-grained statistics**, including distributions of editing task types, text layouts, and font types, and we describe our **sensitive-content filtering strategy** at both the data source and manual screening stages.

> **\[W3\] BBox robustness analysis**

We added a **systematic bbox perturbation experiment**, applying random shifts and scaling of various magnitudes to all bbox corners and tracking metric changes.

Under **random perturbations up to 50%**, the overall and weighted scores remain almost unchanged, showing **strong robustness to bbox noise**.

> **\[W4\] Inference latency benefits**

On IV-Edit, we measured how inference latency changes with the number of regions _K_ for **RePlan vs. multi-round `Flux Kontext`**.

RePlan’s average runtime **increases only slightly** when _K_ grows from 1 to 5, whereas multi-round Kontext’s runtime **grows almost linearly and scales with K**, clearly demonstrating the **efficiency and scalability advantage** of our one-shot multi-region editing.

---

### Author Response · Authors · 2025-12-01
**Rebuttal Summary for the New AC and Reviewers (2/3)**

**2. Generalization**
- **Cross-Model Transfer:** We proved the framework is **plug-and-play** by applying it to the larger `Qwen-Image` backbone, achieving significant gains over the original model _(YTWG, Tvuy, SvSJ)_.
- **External Benchmarks:** We validated generalization by outperforming baselines on established benchmarks like `MagicBrush` and `SmartEdit` _(Tvuy, SvSJ)_.
- **Human Preference:** We conducted a blind **12-expert user study**, where results consistently favored our method over the baseline (`Flux Kontext`) _(YTWG, Tvuy)_.

**3. Clarifications, Statistics & Reproducibility**
- **Benchmark Stats:** We added fine-grained statistics (layout, task type) for the `IV-Edit` benchmark _(LjuJ)_.
- **Implementation Details:** We revised the manuscript with full details on the VLM data pipeline, RL reward formulations, and clearer definitions of "step-by-step" reasoning _(YTWG, Tvuy, SvSJ)_.
- **More Visualization:** We added more comparative visualizations to the appendix and corrected figure references and captions _(Tvuy)_.

- ---

## **4. Summarized Point-by-Point Responses**

For the concerns raised by reviewers, we have provided detailed responses **to each point**.
**We strongly believe that these comprehensive experiments and clarifications solidly resolve the reviewers' concerns:**

---

### **Reviewer YTWG:**

> **\[W1\] Impact of VLM-generated bboxes and detector replacement**

- We **added a systematic bbox perturbation experiment** and found that even when bbox corners are randomly shifted by up to _50%_ of the width/height, the overall and weighted scores remain almost unchanged, with only slight degradation under extreme perturbations. **This shows that our method has strong bbox robustness.**
- We further argue that detectors such as _Grounding DINO_ / _SAM_ **cannot** provide **region-level hints with world knowledge**, and we show that `Qwen2.5-VL` is comparable to Grounding DINO on _RefCOCO_, supporting the choice of a VLM that jointly handles reasoning and localization.

> **\[W2\] Attention behavior for overlapping targets**

We added **visualizations** of overlapping/adjacent cases in the appendix, showing that **RePlan remains stable for multi-region parallel editing**.

When multiple regions overlap, shared patches naturally receive attention from multiple region hints, and we keep the original self-attention among image patches, so we **do not observe extra conflicts or artifacts**.

> **\[W3\] Details of the VLM training data pipeline**

This mainly concerns implementation details of **data sources, filtering, and annotation**. We have clarified these in the response and now provide a **complete description in the appendix** of the revised PDF.

> **\[W4\] Transferability of the attention injection mechanism**

Our method relies on the **joint text–image attention** in MMDiT, but is not tied to a specific model. Replacing `Flux.1 Kontext Dev` with the larger `Qwen-Image`, we find that **`RePlan(Qwen-Image)` still clearly outperforms the original `Qwen-Image`** on all subjective metrics.

This indicates that our **region-level attention injection is general and transferable** within the MMDiT family, and that the overall RePlan framework is **plug-and-play**.

> **\[W5\] Domain diversity of IV-Edit**

IV-Edit currently focuses on **real photos and document editing**, to better evaluate complex multi-target and reasoning-heavy edits. As suggested, we will extend it to more domains to further improve diversity and representativeness.

> **\[W6\] User study and alignment with human preferences**

We added a **user study with 12 experts**, who performed blind comparisons between `Flux Kontext Dev` and `RePlan` on IV-Edit. RePlan is judged **better than Kontext on many more samples than the reverse**, and a portion of cases are considered similar, which further supports our method from the perspective of **human preference**.

> **\[W7\] Unique advantage of VLMs vs. manual bbox + prompt**

- We emphasize that VLMs **remove the need for costly per-image manual bboxes and region prompts**, enabling automatic processing of large-scale editing data, and at inference time they only require **natural-language instructions**, making integration into **agentic systems** straightforward.
- For scenarios with manual refinement, we show on _MagicBrush_ that **using human masks + our attention injection significantly improves all metrics of `Flux Kontext`**, demonstrating that our mechanism can also **seamlessly leverage human annotations**.

---

### Author Response · Authors · 2025-12-01
**Rebuttal Summary for the New AC and Reviewers (1/3)**

Dear new AC and reviewers,

We are deeply grateful for the time and effort you have devoted to reviewing our paper. To respect the new AC’s valuable time and to make it as easy as possible to form an informed judgment, we provide a concise, structured summary below:

(1) Summarized contributions of our work
(2) The main strengths consistently recognized by reviewers
(3) A categorized summary demonstrating the solid resolution of key concerns.
(4) Summarized point-by-point responses to each reviewer

If the new AC has any remaining concerns or further questions regarding the reviewers’ comments, we would be very happy to address them in detail.

---
## **1. Summarized contributions**

We first identify that existing research overlooks **IV-Complexity**, the intricate interplay between complex instructions and visual element, focusing instead on simple, subject-dominated images. To address this, we construct **IV-Edit**, a benchmark dedicated to IV-complex tasks. We then propose **RePlan**, a plug-and-play MMDiT framework enhancing fine-grained perception and reasoning. RePlan couples (1) a lightweight, RL-optimized **VLM Planner** for **region-aligned guidance** with (2) a **training-free attention injection** mechanism. With just **~1k instruction-only samples**, RePlan yields significant overall gains on IV-Edit and external benchmarks both quantitatively and qualitatively, surpassing even closed-source SOTA models in **instance consistency**.

---

## **2. Recognized strengths**

We represent our sincere gratitude for the reviewers' recognition of our work. The consensus on the paper's key strengths is summarized below:

> **1. Clearly defined IV-Complexity and an important IV-Edit benchmark**

The paper systematically defines **Instruction–Visual Complexity (`IV-Complexity`)**, clearly capturing the interaction between **complex instructions** and **complex visuals** (multiple objects, similar instances, cluttered layouts). Reviewers consider this a _“well-defined and important”_ problem setup _(YTWG, SvSJ, LjuJ)_. Building on this formulation, we introduce **`IV-Edit`**, the first benchmark specifically targeting IV-Complex instruction-based editing on real photos, slides, and posters, which reviewers agree **fills an important evaluation gap** _(YTWG, Tvuy, SvSJ, LjuJ)_.

> **2. Region-aligned guidance with training-free region attention injection**

We propose VLM-based **`region-aligned guidance`** and a training-free **`region-level attention injection`** mechanism for `MMDiT`, enabling **precise, parallel multi-region editing without retraining** the DiT, and effectively reducing _edit leakage_ to similar regions. This region-level control is described as **“technically sound, necessary, and interesting”** _(YTWG, Tvuy, SvSJ)_. The planner’s structured outputs further support **interactive and interpretable editing**, which reviewers see as a **practical new interaction paradigm** for instruction-based editing _(YTWG, LjuJ)_.

> **3. Reasoning-driven RePlan with strong IV-Edit performance**

`RePlan` adopts a plan-then-execute design that couples the VLM planner’s **world-knowledge reasoning** and **fine-grained perception** with the diffusion executor’s generative capability, providing a clear and reliable path from **global semantics → region-specific guidance** _(SvSJ, LjuJ, YTWG)_. We use `GRPO` with only `~1k` instruction-only samples, which reviewers find **data-efficient**; ablations show that both RL and CoT reasoning are **necessary components** _(YTWG, LjuJ)_. On `IV-Edit`, we compare against 2 closed-source (e.g., `GPT-4o`, `Gemini`) and 6 open-source models, using `Gemini 2.5-Pro` as a VLM-as-judge over four dimensions: _target localization, consistency, image quality, editing effect_. Reviewers note that **RePlan achieves SOTA among open-source models** and shows clear advantages in **consistency** and **reasoning-grounded editing** under complex instructions, supported by both quantitative metrics and qualitative visualizations _(YTWG, Tvuy, SvSJ, LjuJ)_.

---

## **3. Summarized responses to reviewer concerns**

To facilitate a clear overview, we have summarized the reviewers' concerns into key categories below. **We strongly believe that these comprehensive experiments and clarifications solidly resolve the reviewers' concerns:**


**1. Robustness**
- **BBox Robustness:** We demonstrated strong tolerance to localization noise, showing stable performance even with random bounding box perturbations of up to **50%** _(YTWG, LjuJ)_.
- **Source of Gain:** We verified via ablation that the improvement stems primarily from our **attention injection**, not merely from VLM-refined text prompts _(Tvuy)_.
- **Complex Scenarios:** We added visualizations confirming that our mechanism handles overlapping and adjacent regions without artifacts _(YTWG)_.

---

### Meta-Review · Area_Chair_DNuT · 2025-12-31

**Summary:**

This paper proposes RePlan, a reasoning-guided region planning framework for complex instruction-based image editing, which introduces the IV-Complexity definition and the IV-Edit benchmark, and achieves performance improvements through training-free region attention injection. Reviewers generally recognize the work's strengths, including the clear definition of IV-Complexity that captures the interaction between complex instructions and visual scenes, the innovative region-aligned guidance mechanism, and the valuable IV-Edit benchmark for evaluating complex image editing tasks.

Most technical concerns raised by reviewers are mostly resolved through detailed experiments, supplementary analyses, and manuscript revisions. Inquiries regarding the impact of VLM-generated bboxes, attention behavior for overlapping targets, VLM training data pipeline, transferability of the attention injection mechanism, user study alignment with human preferences, and the unique advantages of VLMs over manual bbox prompts are all effectively addressed. Concerns about unclear references / descriptions of Figures 9-12, the source of improvements, RL reward design and label format, lack of metrics / benchmark comparisons, and insufficient visual results are fully resolved. For questions about the ambiguity of “step-by-step reasoning”, limited quantitative gains, and incomplete RL details, the authors have provided satisfactory responses. Concerns regarding evaluation bias in ablation studies, insufficient dataset statistics, bbox robustness, and inference latency are also mostly addressed.

However, some concerns are not sufficiently resolved despite supplementary explanations. First, Reviewer YTWG's suggestion on expanding the domain diversity of IV-Edit: the authors state plans for future expansion but fail to provide specific implementation details or preliminary results, leaving the current benchmark's limited domain coverage unaddressed. Second, Reviewer Svsj's question about whether IV-Edit is sufficient for assessing reasoning generalization: the authors only note that the dataset scale is comparable to existing benchmarks (e.g., MagicBrush, SmartEdit) but do not adequately demonstrate IV-Edit's ability to comprehensively evaluate reasoning generalization across diverse scenarios. Third, Reviewer Tvuy's doubt about Qwen-Image achieving higher weighted scores: while the authors highlight RePlan's advantages in other metrics (e.g., quality, consistency), they do not directly explain why Qwen-Image outperforms in weighted scores or prove RePlan's absolute superiority in overall performance.

Based on the above considerations, I think the current manuscript does not match the ICLR’s requirement and I do not recommend to accept this paper.

**Reviewer Concerns:**

Reviewer YTWG's suggestion on expanding the domain diversity of IV-Edit: the authors state plans for future expansion but fail to provide specific implementation details or preliminary results, leaving the current benchmark's limited domain coverage unaddressed. Reviewer Svsj's question about whether IV-Edit is sufficient for assessing reasoning generalization: the authors only note that the dataset scale is comparable to existing benchmarks (e.g., MagicBrush, SmartEdit) but do not adequately demonstrate IV-Edit's ability to comprehensively evaluate reasoning generalization across diverse scenarios. Reviewer Tvuy's doubt about Qwen-Image achieving higher weighted scores: while the authors highlight RePlan's advantages in other metrics (e.g., quality, consistency), they do not directly explain why Qwen-Image outperforms in weighted scores or prove RePlan's absolute superiority in overall performance.

**Reviewer Scores:**

Reviewers would keep their scores.

---

### Decision · Program_Chairs · 2026-01-26

Reject